# Months-long tracking of neuronal ensembles spanning multiple brain areas with Ultra-Flexible Tentacle Electrodes

Tansel Baran Yasar [1,2,5], Peter Gombkoto[1,2,5], Alexei L. Vyssotski [1,2], Angeliki D. Vavladeli[1,2], Christopher M. Lewis [2,3], Bifeng Wu[1,2], Linus Meienberg [1], Valter Lundegardh[1], Fritjof Helmchen [2,3,4], Wolfger von der Behrens [1,2] & Mehmet Fatih Yanik [1,2] ✉

We introduce Ultra-Flexible Tentacle Electrodes (UFTEs), packing many independent fibers with the smallest possible footprint without limitation in recording depth using a combination of mechanical and chemical tethering for insertion. We demonstrate a scheme to implant UFTEs simultaneously into many brain areas at arbitrary locations without angle-of-insertion limitations, and a 512-channel wireless logger. Immunostaining reveals no detectable chronic tissue damage even after several months. Mean spike signal-to-noise ratios are 1.5-3x compared to the state-of-the-art, while the highest signal-to-noise ratios reach 89, and average cortical unit yields are ~1.75/channel. UFTEs can track the same neurons across sessions for at least 10 months (longest duration tested). We tracked inter- and intra-areal neuronal ensembles (neurons repeatedly co-activated within 25 ms) simultaneously from hippocampus, retrosplenial cortex, and medial prefrontal cortex in freely moving rodents. Average ensemble lifetimes were shorter than the durations over which we can track individual neurons. We identify two distinct classes of ensembles. Those tuned to sharp-wave ripples display the shortest lifetimes, and the ensemble members are mostly hippocampal. Yet, inter-areal ensembles with members from both hippocampus and cortex have weak tuning to sharp wave ripples, and some have unusual months-long lifetimes. Such inter-areal ensembles occasionally remain inactive for weeks before re-emerging.

Complex behaviors such as perception, decision-making, and learning involve the interaction of many brain areas[1,2]. Techniques to perform long-term recordings of brain activity at single-cell resolution simultaneously from distributed brain areas are essential to investigate the coordinated brain dynamics underlying the learning and execution of such behaviors. Such high-resolution recordings can also improve our understanding of brain disorders and enable the development of new diagnostics and treatments that would be otherwise combinatorially impossible to discover using behavioral readouts alone[3]. However, currently available technologies for such recordings are limited, both in terms of their long-term stability and their applicability to multi-areal recordings. Although optical methods enable stable recordings of single-cell activity[4], they cannot reach deep brain areas such as the hippocampus or thalamus without damaging the intermediate brain

[1]Institute of Neuroinformatics, ETH Zurich & University of Zurich, Zurich, Switzerland. [2]Neuroscience Center Zurich, University of Zurich & ETH Zurich, Zurich, Switzerland. [3]Brain Research Institute, University of Zurich, Zurich, Switzerland. [4]University Research Priority Program (URPP), Adaptive Brain Circuits in Development and Learning, University of Zurich, Zurich, Switzerland. [5]These authors contributed equally: Tansel Baran Yasar, Peter Gombkoto. ✉e-mail: yanik@ethz.ch

areas and require compromises between the number of cells recorded and temporal resolution[5]. They also require genetic modifications or the delivery of fluorescent indicators to neurons, making them impractical to use in the human brain.

Extracellular electrode arrays, on the other hand, are typically fabricated on stiff materials and with large footprints that do not comply with the brain tissue, leading to glial encapsulation of the electrode arrays, loss of single units, low unit yields, drifts in recordings, and the inability to track single units across sessions[6–10], unless ultra-high-density contacts are used to compensate for drifts and instabilities[11,12]. Flexible polymer electrode arrays have emerged as alternatives to stiff probes, offering superior biocompatibility, adaptability to the mechanics of the brain tissue, and stability of recordings[13–17]. However, recording from multiple brain areas with high-density flexible electrode arrays at cellular scale (<10 μm) without limitations on implantation depth while minimizing the implants' footprints still remains a major challenge. Integrating more recording contacts on polymer shanks requires them to be wider, which leads to cutting through more neuronal structures during insertion (Fig. 1a). While this can be mitigated by distributing the recording contacts into independent tiny electrode fibers, inserting many such fibers simultaneously has been challenging due to their high flexibility and the shortcomings of the chemical glues to keep them together (Fig. 1b). For instance, polyethylene glycol, a water-soluble material commonly used for tethering electrode arrays to shuttles, can be dissolved in seconds inside the brain or even near the brain surface due to the humidity. Achieving stiffness or stability for insertion by using even

more adhesive alone leads to larger insertion footprints. If, instead, a stiff insertion shuttle is used to achieve stiffness while holding together all the fibers and shuttle using a strong glue, the electrode fibers often get pulled out during retraction of the stiff insertion shuttle (Fig. 1b).

To overcome these issues, we developed Ultra-Flexible Tentacle Electrode (UFTE) arrays, which provide stable recordings of single units with exceptionally high mean signal-to-noise ratios (SNRs) of single-unit spikes, 1.5–3 times larger than state-of-the-art flexible electrode arrays[15,16], with some mean single-unit SNRs as large as 89. We demonstrate that UFTEs can be inserted at least 6.5 mm deep from the dorsal surface while achieving high single-unit yields per recording contact. We also developed a technique to insert UFTEs simultaneously into many brain areas at arbitrary locations without any depth and angle-of-insertion limitations (Fig. 2d, e) to vastly simplify distributed recordings with UFTEs.

Our approach uses thin, mechanically uncoupled, ultra-flexible polyimide fibers that self-assemble into bundles to efficiently pack many recording contacts in a small footprint (Figs. 1c and 2a). Fibers are held together with a biodegradable glue during insertion into the brain. However, unlike other approaches (Fig. 1b), ours does not rely solely on the tissue-dependent dissolution kinetics of biodegradable glues. The silk glue holds only the electrode fibers together for each bundle but is not used to connect the bundle to the insertion shuttle. Instead, we mechanically couple the bundle of many fibers to the insertion shuttle via a loop at the tip of only one fiber in each bundle (Figs. 1c and 2a, c). Mechanical coupling to a thin yet stiff insertion shuttle enables reliable insertion of the electrode bundle with many

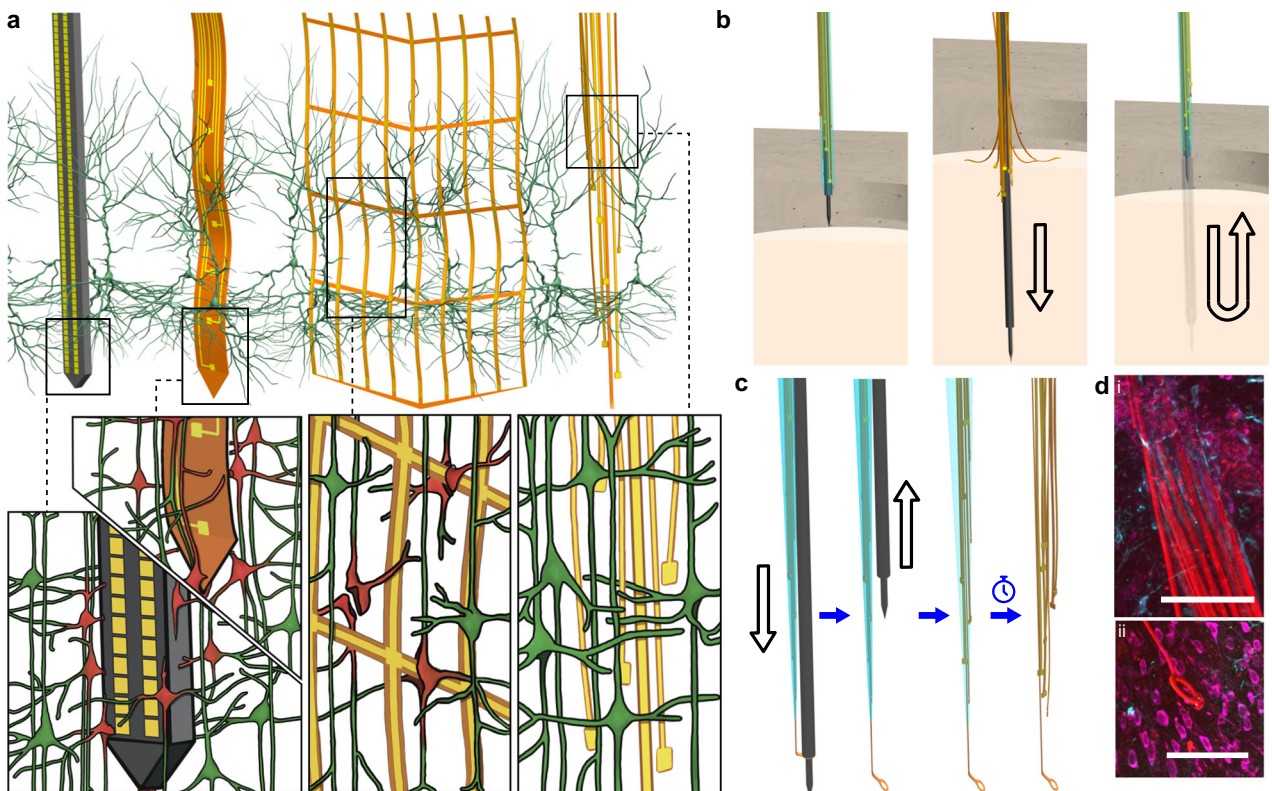

**Fig. 1 | Ultra-Flexible Tentacle Electrodes (UFTEs) with cellular-scale footprint reduce tissue damage and are implantable to deep brain regions. a** Drawn-to-scale comparison of the geometries of a rigid silicon probe, flexible planar shank probe, neural mesh, and UFTE (from left to right) with reference to the soma and dendrites of surrounding neurons. Zoomed insets (bottom) show the impact of different probe geometries on neuronal processes after insertion into the brain.
**b** Potential failures during the insertion of ultra-flexible electrode arrays coupled to a stiff shuttle purely by chemical tethering (left). The electrode fibers can separate

prematurely from the stiff shuttle if the coating dissolves too fast (middle), or they can get stuck to the shuttle and be pulled out of the brain during shuttle retraction if the coating dissolves too slowly (right). **c** Delivery of the UFTE bundle into the brain, where the bundle is inserted into the brain with the help of a tungsten shuttle, the shuttle is retracted, and the silk coating is dissolved, leaving the electrode fibers independent from each other. **d** UFTE 3.5 months post-implantation. (i) UFTE bundle (red) shown with stained neurons (magenta) and the microglia (cyan). (ii) Loop of UFTE bundle in the same brain slice. Scale bars are 100 μm.

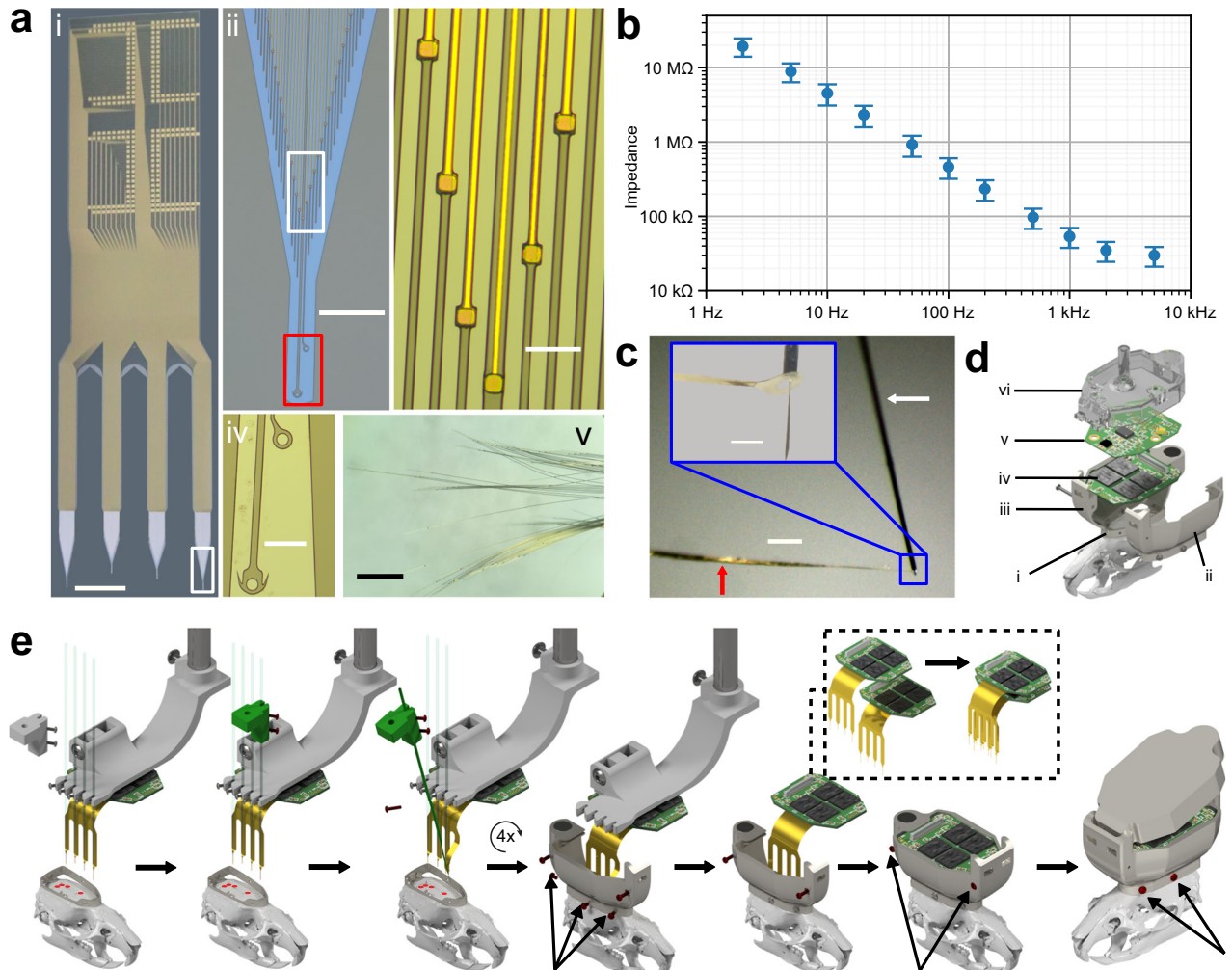

**Fig. 2 | Fabrication and assembly of UFTEs with multi-areal implantation system. a** (i) 4-bundle, 256-channel UFTE (ii) Close-up view of the white rectangle in (i), where the individual electrode and loop fibers are visible. (iii) Enlarged view of white rectangle in (ii). Polyimide-encapsulated gold wire tracks in each electrode fiber are terminated with a PEDOT:PSS-coated gold contact pad. (iv) Enlarged view of red rectangle in (iii), where primary and auxiliary loops are visible. (v) Electrode fibers of a bundle floating in distilled water. Scale bars: 5 mm (i), 500 µm (ii), 100 µm (iii) and (iv), 1 mm (v). **b** Impedance spectroscopy of a 256-channel UFTE in Ringer's solution pre-assembly (mean ± s.d., $n = 243$ channels). Broken channels (impedances at 1 kHz > 5 MΩ, $n = 4$) and channels that are not coated with PEDOT:PSS (impedances at 1 kHz with z-scores > 3 among non-broken channels, $n = 9$) are excluded as outliers. **c** Tip of tungsten shuttle is inserted into the loop of silk-coated electrode bundle during the assembly process. Red and white arrows show silk-coated electrode bundle and tungsten shuttle. Inset with blue outline shows zoomed-in view of the tip of tungsten shuttle with an electrode loop attached. Scale bars: 500 µm and 100 µm (inset). **d** Exploded view of the components of Tita-niumHelmet, 256-channel headstage, and recording cap illustrated on rat skull model: (i) base, (ii) and (iii) left and right shells, (iv) custom 256-channel headstage, (v) recording board with LEDs, head-orientation sensor, and connections to the commutator (vi) 3D-printed recording cap. **e** Steps of the implantation process for each bundle of electrodes. Moving parts in each step are highlighted in green. Screws inserted in each step are highlighted with red color and arrows. The implantation process can be adapted to different numbers of bundles, insertion angles/depths/locations. Multiple head stages can be stacked on top of each other. Source data are provided as a Source Data file.

independent fibers into the brain (Fig. 1c) and the immediate removal of the insertion shuttle without waiting for the glue holding the fibers to dissolve. Due to the reliance on mechanical insertion, stronger glues can be used, thereby permitting controlled insertion into deep brain regions.

Our sub-cellular-sized self-assembling electrode fibers (Fig. 1d) achieve smaller implant footprints compared to other approaches using wider flexible shanks to achieve high-channel counts. For the same recording contact sizes, a flexible 32-channel planar shank with 100 µm width at the base[15] and the ultra-flexible mesh electrodes inserted in an open formation[16] would occupy 2.5 times and 22 times larger cross-sectional area per recording contact in the brain tissue than UFTEs during the electrode implantation (accounting for the space filled with biodegradable glues and shuttles), respectively

(Fig. 1a). The small size and ultra-flexibility of the UFTE fibers remaining in the brain after the removal of the insertion shuttle allow them to interdigitate with the neuronal structures, resulting in months-long stable recordings of single units.

The hippocampus, retrosplenial cortex, and medial prefrontal cortex are strongly connected brain regions crucial for memory, navigation, inference, and decision-making[18–23]. However, simultaneous interactions of these brain areas as a network and the long-term dynamics of these networks remain unexplored. Here, we used UFTEs to observe these interactions while simultaneously recording from these brain areas for months-long periods. In particular, we captured the dynamics of neuronal ensembles (neurons co-activated within 25 ms) spanning multiple brain areas for 3.5 months and their interactions with sharp wave ripples (SWRs) in freely moving rats.

## Results

### Fabrication and assembly of the Ultra-Flexible Tentacle Electrodes

We fabricated UFTEs with 256 recording contacts (Fig. 2a; see Methods and Supplementary Fig. 1 for further details of the microfabrication process). The electrode arrays consisted of a Ti/Au metal layer sandwiched between two polyimide layers, which has excellent long-term stability in physiological conditions[24]. The electrode arrays had a unique geometry with one recording contact per fiber, each completely mechanically decoupled from one another (Fig. 2a and Supplementary Fig. 8). Each fiber was 7 μm wide and 2.4 μm thick, resulting in a cross-sectional area of 16.8 μm$^2$. This is one order of magnitude smaller than the cross-sectional area of the soma of a typical neuron[25]. The recording contacts had a surface area of $13 \times 13$ μm$^2$ and were coated with PEDOT:PSS, which reduced the electrode-electrolyte impedance magnitude to $54 \pm 16$ kΩ (mean ± s.d., $n = 243$ recording contacts) at 1 kHz (Fig. 2b, impedance phases are provided in Supplementary Fig. 7a). The percentage of broken channels in the in vitro impedance measurements was -1.6%, which can be attributed to defects in the microfabrication or soldering processes. The percentage of recording contacts that we excluded from our spike sorting analysis due to not recording any local field potential or spiking activity was 3–6%.

The 256 recording contacts were distributed over four bundles with 64 recording contacts per bundle to enable distributed recordings from different brain areas at arbitrary distances from each other. We terminated the longest fiber in the electrode bundle with a loop of 25 μm inner diameter (Fig. 2a), which we used to mechanically tether the bundle to a tungsten shuttle (Fig. 2c). The other fibers in the bundle extended 500 μm past their recording contacts to achieve a streamlined conical shape of the UFTE bundles and to reduce the risk of UFTE fibers separating from each other prematurely during insertion.

After fabrication, we peeled the UFTEs off the wafer and soldered them onto custom headstages (Supplementary Fig. 2a). Peeling the electrode fibers off the wafer while they were immersed in water resulted in the self-assembly of the electrode fibers into bundles. We coated each bundle with PEG solution and silk fibroin (Supplementary Fig. 2b) to keep them together during implantation and inserted the tip of a tungsten shuttle into the loop at the end of each electrode bundle (Fig. 2c). Through this process, we assembled each of the four bundles with a respective tungsten shuttle and held all these components in a 3D-printed electrode holder (Supplementary Fig. 2c).

### High-density UFTE recordings from multiple brain areas in freely moving rats

We implanted four UFTE bundles spanning six different brain regions: the medial prefrontal cortex (mPFC), including infralimbic (IL) and prelimbic (PrL) cortices and the cingulate (Cg1), retrosplenial cortex (RSC), dorsal hippocampus (dHPC), and intermediate hippocampus (iHPC). Before inserting each bundle, we transferred its tungsten shuttle from the 3D-printed electrode holder to a stereotaxic arm and inserted the shuttle into the brain at 12.5 μm/s speed (Fig. 2e). The flexibility of the polyimide cable between the electrode bundle and the headstage allows the user to insert electrode bundles into arbitrary locations with wide ranges of insertion angles. We implanted a total of 13 electrode bundles into the targeted brain areas in 4 rats. Furthermore, due to the mechanical tethering between the electrode bundles and the tungsten shuttles, we were able to push the electrode bundles through the dura mater and minimally disturb the protective meninges. The lateral spread of the UFTE fibers in the brain tissue was estimated at around 100 μm, according to post-mortem images of the tissue (Fig. 1d).

After a post-surgery recovery period (2–6 days), we recorded local field potentials (LFP) and spiking activities from all six brain areas of freely moving rats twice a week in a 50 cm×50 cm clear acrylic cage,

with signals matching their established neurophysiological characteristics (Fig. 3 and Supplementary Video 1). We recorded SWRs and single units putatively corresponding to pyramidal cells from the hippocampal electrode bundles. We detected units with either intrinsic theta rhythm[26] or tuning to the head direction angle[27] from the electrode bundles in RSC.

In one of the recording sessions, we detected a single unit with a large-amplitude positive waveform ($441.4 \pm 1.3$ μV amplitude, mean ± s.e.m., $n = 755$ spikes) from a recording contact in the RSC (Fig. 3a). The background activity in this recording contact had a reverse polarity compared to the LFP in the neighboring contacts during UP and DOWN state transitions, potentially indicating a cell-attached recording configuration. We observed several single units with high SNRs in other recording sessions and brain areas as well, such as single units with 89.2 SNR ($835.2 \pm 13.6$ μV amplitude, $n = 17$ spikes) and 68.6 SNR ($623.6 \pm 2.6$ μV amplitude, $n = 833$ spikes) in dHPC, 63.2 SNR ($412.3 \pm 3.0$ μV amplitude, $n = 316$ spikes) in iHPC, and 63.1 SNR ($535.5 \pm 0.3$ μV amplitude, $n = 12282$ spikes) in mPFC (mean ± s.e.m. is reported for the amplitudes of all units).

Laminar profiles of the LFPs recorded by the hippocampal UFTE bundles during SWRs were similar to the spatial LFP patterns recorded previously by stiff probes[18]. Furthermore, we observed a similar laminar order in the UFTE bundle implanted into the mPFC during slow oscillations of NREM sleep. These observations indicate that the relative spatial order of the recording contacts is preserved after the UFTE bundles are inserted into the brain.

### Long-term stability and biocompatibility of UFTE recordings

To assess the biocompatibility and the stability of single-unit recordings, we implanted UFTEs into two rats for 3.5 months and performed recordings twice a week. We were able to easily track single units across sessions (Fig. 4a). Some single units were tracked for months, as verified by the spike waveform profile of the units across multiple channels and the firing characteristics (Fig. 4b).

The impedance values of the recording contacts remained stable across 3.5 months (Fig. 4c). Based on the weekly impedance measurements, the impedance values started higher than the in vitro values, slightly increased during the first month after implantation and stabilized afterwards. For Rat #1 (Fig. 4c, red), the median of the impedance magnitudes measured at 1 kHz on post-implantation days 27, 61 and 95 were 257.5 kΩ (Interquartile Range (IQR) = 296.5 kΩ, $n = 184$), 262.0 kΩ (IQR = 453.0 kΩ, $n = 181$), and 313.0 kΩ (IQR = 437.0 kΩ, $n = 185$), respectively. For Rat #2 (Fig. 4c, blue), the median of the impedance magnitudes measured at 1 kHz on post-implantation days 24, 59, and 90 were 329 kΩ (IQR = 498.5 kΩ, $n = 236$), 450.5 kΩ (IQR = 988.3 kΩ, $n = 234$), and 540.5 kΩ (IQR = 1224 kΩ, $n = 230$), respectively ($n$ = number of functional recording contacts in all cases; impedance phases across 3.5 months are reported in Supplementary Fig. 7b).

To track single units across sessions, we took 20-minute excerpts from each recording session and concatenated these for each rat. We then detected and sorted the spikes into clusters and identified each cluster as single- or multi-unit in a semi-automatic manner (GPU-based automatic clustering[28] followed by manual curation). Alongside expert judgment on cluster separation from noise and neighboring clusters during the manual sorting process, we eliminated clusters with more than 2% interspike intervals (ISI) violating a 2 ms threshold[29]. We identified three quality metrics for each cluster to quantify manual sorting performance: SNR, percentage of ISI violations (2 ms threshold), and Mahalanobis distance from the nearest neighbor. For all these metrics, there was a significant difference between single and multi-unit clusters (Supplementary Fig. 4, SNR: $14.6 \pm 0.5$ vs. $6.8 \pm 0.1$ for single- vs. multi-units, mean ± s.e.m.; median = 12.1, IQR = 9.4 for single- vs. median = 6.1, IQR = 2.9 multi-units; $p = 1.26 \times 10^{-64}$ with Wilcoxon's Rank-Sum test; Percentage of ISI violations: median = 0.2,

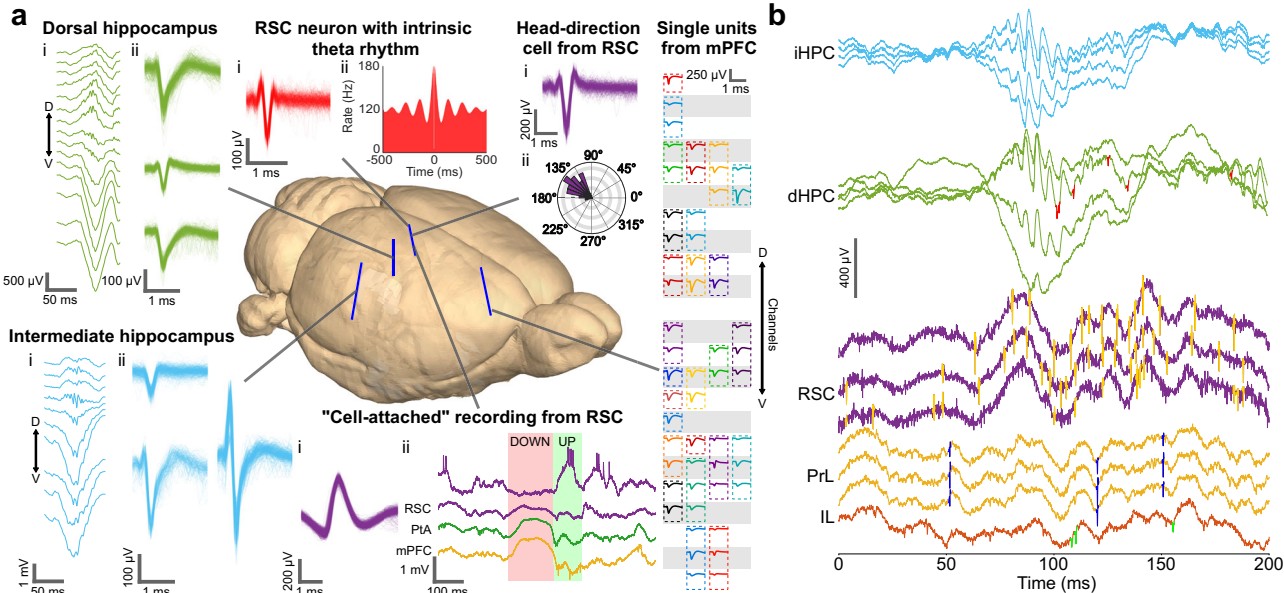

**Fig. 3 | Simultaneous high-density recordings from six brain areas of freely moving rats with four UFTE bundles with preserved laminar distributions.**
**a** Laminarly-resolved single-unit recordings and local field potential (LFP) recorded by UFTEs. Drawn-to-scale schematic of rat brain (generated by Waxholm Rat Atlas[87]) showing electrode locations (blue). *Dorsal hippocampus (dHPC)* and *Intermediate hippocampus (iHPC):* (i) Laminar distribution of LFP during sharp-wave ripple. Rows are signals from neighboring channels. Arrow shows the dorsoventral axis ("D": dorsal, "V": ventral). (ii) Waveforms of 3 sample single units detected in the CA3 pyramidal layer of dHPC or iHPC. *RSC (Retrosplenial cortex) neuron with intrinsic theta rhythm:* (i) Mean single unit waveform. (ii) Autocorrelogram of spikes of this single unit showing an oscillation at theta rhythm. *Head-direction cell from RSC.* (i) Mean single unit waveform. (ii) Polar plot shows tuning of this single unit to an angle of -135°. *Single units from medial prefrontal cortex (mPFC) (PrL, IL, Cg1):*

Mean waveforms of single units detected in mPFC. Each colored box corresponds to one single unit. Each gray/white row represents a recording contact on the UFTE bundle. Arrow shows dorsoventral axis. *Cell-attached recording from RSC:* (i) Spike waveforms of the single unit. (ii) Raw data showing the behavior of the "cell-attached neuron" during down- to up-state transition during sleep (top) with simultaneous raw data from RSC, PtA (parietal association cortex), and mPFC (bottom). **b** Simultaneous temporal recording traces from multiple brain areas during SWR in hippocampus. 200-ms simultaneous recording from four recording contacts in iHPC (light blue), four recording contacts in dHPC (green), three recording contacts in RSC (purple), three recording contacts in PrL (yellow), and one recording contact in IL (orange). Neuronal spikes are highlighted by red (dHPC), orange (RSC), dark blue (PrL), and green (IL). Source data are provided as a Source Data file.

IQR = 0.5 for single- vs. median = 1.4, IQR = 1.6 multi-units, $p = 6.55 \times 10^{-67}$ with Wilcoxon's Rank-Sum test; Distance from the nearest neighbor: median = 3.02, IQR = 1.75 for single- vs. median = 2.03, IQR = 0.76 for multi-units, $p = 2.46 \times 10^{-49}$ with Wilcoxon's Rank-Sum test. Number of single-units = 445, number of multi-units = 345).

To investigate whether the single units continuously detected by the semi-automatic clustering across sessions belonged to the same neuron, we split the spike trains of the identified clusters back into 20-min time blocks corresponding to individual recording sessions. We calculated the mean waveforms on the recording contacts for each recording session and unit. Using principal component analysis, we reduced the dimensionality of each spike waveform in the unit from [64 recording contacts × 41 samples] to [64 recording contacts × 3 principal components].

For each unit, we calculated two metrics for the spikes from the tracked unit versus spikes from different units: Pearson's correlation coefficient between mean waveforms and standardized mean difference (SMD) between clusters in the PCA space. We calculated these metrics for three categories of unit/time block pairs: (i) spikes belonging to the same unit across pairs of different time blocks to assess unit stability over long-term ("same-unit pairs"), (ii) spikes belonging to different units than the unit of interest across all time blocks ("different-unit pairs"), (iii) spikes belonging to different units than the unit of interest but within the spatial vicinity of ±3 recording contacts from the center recording contact of the unit of interest ("neighboring different-unit pairs"). The Pearson's correlation coefficient between the mean spike waveforms of groups of "same-unit pairs," "different-unit pairs," and "neighboring different-unit pairs" were 0.98 (0.04), −0.0002 (0.02), 0.33 (0.49) respectively

(Supplementary Fig. 5a). The SMD between the spike clusters of "same-unit pairs," "different-unit pairs," and "neighboring different-unit pairs" were 1.27 (1.15), 11.25 (4.62), 8.21 (4.05), respectively (Supplementary Fig. 5b). (median (IQR) is reported in all cases; n = 5225 for "same-unit pairs", n = 3501090 for "different-unit pairs", n = 741754 for "neighboring different-unit pairs"). Pearson's correlation coefficient and the SMD for the "same-unit pairs" were significantly lower than those of the "different-unit pairs" and "neighboring different-unit pairs" ($p < 10^{-100}$ for all comparisons using Wilcoxon Rank-Sum Test). Furthermore, these metrics were significantly lower for the "same-unit pairs" between the different time blocks of each unit than the "neighboring different-unit pairs" in every unit we tracked longitudinally (except the cases where there was only one "same-unit pair").

We quantified the number, quality, and longevity of the single units detected in each rat across 3.5 months. The SNR of all single units was 11.1 ± 0.6 (n = 95 SU) during the first week, 13.5 ± 0.4 (n = 377 SU) after one month, 12.7 ± 0.3 (n = 486 SU) after two months, and 11.2 ± 0.3 (n = 410 SU) after three months after the electrode implantation (all values are reported as mean ± s.e.m.; Fig. 4d). The unit yield per recording contact in the cortical bundles was 1.41 during the first week (251 units), 1.62 at one month (289 units), 1.76 at two months (313 units), and 1.50 at three months (267 units; per 178 recording contacts in all cases) after the electrode implantation. The unit yield per recording contact in the hippocampal bundles was 0.93 during the first week (232 units), 0.97 at one month (241 units), 0.85 at two months (211 units), and 0.80 at three months (200 units; per 249 recording contacts in all cases) after the electrode implantation (Fig. 4d). 19.1% of all the recorded single units were trackable for three months (Fig. 4e).

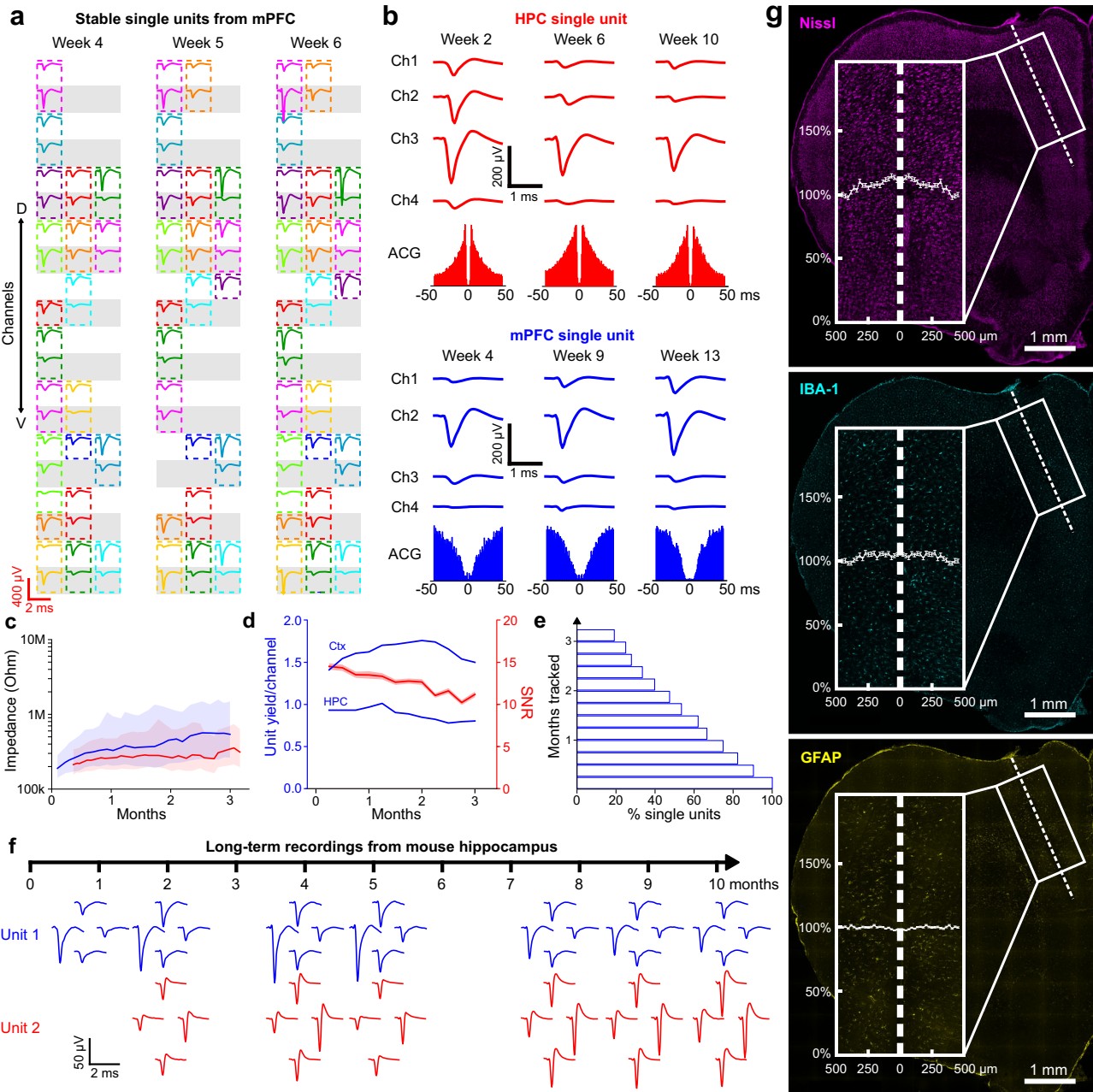

**Fig. 4 | UFTE single-units are stable up to almost a year (10 months, longest tested), with no detectable adverse tissue response around electrodes after months. a** Mean single unit waveforms from mPFC (medial prefrontal cortex) across three recording sessions one week apart. Each colored box corresponds to one single unit. Each gray/white row represents a recording contact on electrode bundle (arrow shows dorsoventral axis). **b** Sample HPC (hippocampus) single unit tracked for two months (top, red), and sample mPFC single unit tracked for nine weeks (bottom, blue). Mean waveforms across four neighboring recording contacts and auto-correlograms (ACG) are shown for each unit. **c** Electrode-tissue impedance magnitudes during 3.5 months in two rats at 1 kHz frequency (red: Rat #1, blue: Rat #2). For each data trace, line shows median, filled area shows interquartile range across functional recording contacts ($n = 184, 181, 185$ contacts on days 27, 61, 95 for Rat #1 and $n = 236, 234, 230$ contacts on days 24, 59, 90 for Rat #2). **d** Unit yield per recording contact (blue) from cortical bundles ("Ctx") and hippocampal bundles ("HPC") from two rats across 3.5 months. Signal-to-noise ratio (SNR, red) of single units recorded from two rats across 3.5 months (Mean ± s.e.m.; $n = 95, 377, 486, 410$ single units at Week 1, 4, 8, 12). **e** Longitudinal tracking durations of single units recorded from the two rats. **f** Single units from mouse hippocampus tracked for 10 months (longest duration tested). Spike waveforms from four recording contacts on tetrodes are shown. **g** Immunohistology of brain slice showing tissue implanted with UFTE bundle in mPFC, 3.5 months post-implantation. Stainings were done for Nissl (magenta), IBA-1 (cyan), and GFAP (yellow). Insets show zoomed-in view of white boxes, accompanied by fluorescence intensity relative to that at 500 μm distance from center of bundle. Graphs in insets show mean ± s.e.m. Dashed lines show where the electrode bundles used to be. The images are maximum projections of image stacks acquired by confocal microscopy. Source data are provided as a Source Data file.

Furthermore, 39.8%, 66.5%, and 90.3% of all single units were trackable for more than 2 months, 1 month, and 1 week, respectively (The total number of single units was $n = 445$ across all recording contacts in the two rats).

A variant of UFTEs where four recording contacts were placed per polyimide fiber in tetrode configuration was implanted in the CA1 region of the hippocampus of mice ($n = 2$). During these experiments, which were conducted for almost a year (10 months; longest duration

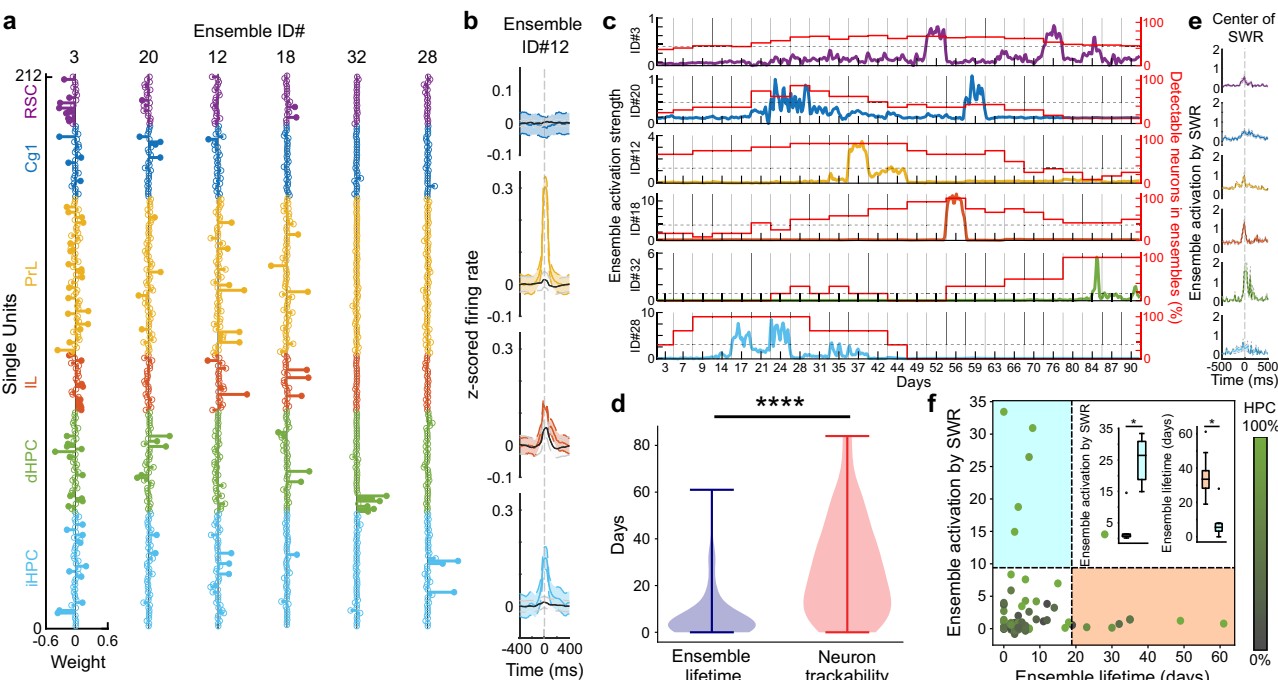

**Fig. 5 | Inter- and intra-areal hippocampal and cortical neuronal ensembles tracked for months. a** Vertical columns display weight distributions (x-axis) of six ensembles constructed from neurons (y-axis) originating from RSC (retrosplenial cortex, purple), CG1 (cingulate cortex, cyan), PrL (prelimbic cortex, yellow), IL (infralimbic cortex, orange), dHPC (dorsal hippocampus, green), and iHPC (intermediate hippocampus, blue). Filled and non-filled dots represent member and non-member neurons of the ensemble. **b** Comparison of z-scored firing rates between ensemble members and non-members in Ensemble #12, triggered by time points where ensemble activation strengths exceed 2 x s.d., shows significant differences for PrL, IL, and iHPC member neurons during ensemble activation ($n = 20028$ ensemble activation events, $p = 9.56 \times 10^{-8}$, $9.33 \times 10^{-8}$, $9.09 \times 10^{-10}$ for PrL, IL and iHPC, two-sided Student's $t$ test). **c** Ensemble activation strength (left y-axis, dashed black line: +2 x s.d.) and the percentage of detectable member neurons (right y-axis, red) over three-months. **d** Comparison of trackability of individual neurons (red) with lifetimes of ensembles (blue) ($n = 79$ ensembles, $p = 1.41 \times 10^{-10}$, two-sided Wilcoxon rank-sum test). Lines on violin plots show minima/maxima. **e** Sharp-wave

ripple (SWR)-triggered ensemble activation strength corresponding to ensembles presented in **c**. **f** Distributions of ensembles based on their activation strengths during SWRs and their lifetimes. Color scale (HPC) indicates percentage of hippocampal neurons in ensembles. Horizontal and vertical black dashed lines show mean + s.d. of activation strengths during SWR and mean + s.d. of ensemble lifetimes, delineating two groups: short-lived ensembles with high activation strength (cyan area) and ensembles with long lifetimes but low activation strength (orange area). Left inset shows difference in activation strength during SWRs between short-lived (green, $n = 5$) and long-lived (gray, $n = 9$) ensembles ($p = 0.0027$, two-sided Wilcoxon's rank-sum test). Right inset shows difference in ensemble lifetimes between ensembles with high (green, $n = 6$) and low activation strengths (gray, $n = 8$) ($p = 0.0045$, two-sided Wilcoxon's rank-sum test). In box plots, center line shows median, bounds show lower and upper quartiles, whiskers show (lower quartile − 1.5 × interquartile range) and (upper quartile + 1.5 × interquartile range), fliers show values outside range shown by whiskers. Source data are provided as a Source Data file.

tested), single units were detectable until the termination of the experiment. The mean single-unit SNRs and electrode impedances at 1 kHz were also stable (Supplementary Fig. 9). Some single units were tracked across months based on the profile of the mean spike waveforms on the recording contacts of the tetrodes (Supplementary Fig. 9). For example, two of the single units (Fig. 4f) were trackable for 277 (Unit #1) and 257 (Unit #2) days.

The histological analysis of the brain tissue around the implants in rats showed no detectable adverse chronic reaction after 3.5 months of implantation (Fig. 4g). We stained the rat brain slices that used to contain electrode bundles to quantify neurons around the implant area (Neurotrace 640/660) and to evaluate the immune response in the form of microglia (IBA-1) and activated astrocytes (GFAP). In one sample slice, there was a 10.9% ± 1.5% increase in the density of neurons in the direct vicinity of the electrode tract compared to further away (500 μm distance). There was an insignificant 5.1% ± 1.4% and 0.8% ± 0.4% increase in the vicinity of the implant site in the biomarkers of microglia and activated astrocytes compared to 500 μm away, respectively. (Kolmogorov-Smirnov Test; $p = 0.33$ and $p = 0.99$ for microglia and astrocytes, respectively; all values above are reported as mean ± s.e.m.) Other sample brain slices with immunostaining are shown in Supplementary Fig. 6.

## Hippocampal ensembles with low SWR-tuning remained detectable longer across months

UFTEs enabled continuous monitoring of neurons and intra- and interareal ensemble activities for over three months in the rat brain. Our analysis focused on identifying neuronal ensembles that can be defined as a group of neurons displaying repeated coincident firing patterns within a narrow time window (25 ms). Additionally, we investigated the stability of neuronal ensembles and whether their lifetimes exhibited variations across all recording sessions. We detected 34 (Rat #1) and 45 (Rat #2) ensembles. We determined their weight vectors, representing each neuron's contribution to a given ensemble (Fig. 5a, see Methods for the details and parameters of ensemble detection[30]). These ensembles comprised neurons from diverse brain regions, including PrL, IL, Cg1, RSC, and various hippocampal regions. Eleven of these ensembles primarily consisted of neurons from hippocampal regions, while three were predominantly composed of neurons from PrL, and the remaining ensembles exhibited mixed neuron compositions. To confirm alterations in the firing rates of ensemble members across various brain structures, we compared the z-scored firing rates of neuronal members and non-members within the same ensemble triggered by ensemble activations. Our analysis revealed significant changes in the firing rates of ensemble members compared to non-members within regions such as PrL (z-scored firing

rate of non-member: 0.01 ± 0.02; member: 0.25 ± 0.03, mean ± s.d., Student's $t$ test; $p = 9.56 \times 10^{-8}$), IL (non-member: 0.04 ± 0.03; member: 0.10 ± 0.03, mean ± s.d., Student's $t$ test; $p = 9.33 \times 10^{-8}$), the intermediate hippocampus (non-member: 0.01 ± 0.02; member: 0.14 ± 0.03, mean ± s.d., Student's $t$ test; $p = 9.09 \times 10^{-10}$, Fig. 5b).

Additionally, we investigated whether the lifetimes of ensembles were limited due to the detectability of neurons or the changes in the temporal correlations among neurons over time. In some sessions, even when all the neurons in an ensemble were detectable, the ensemble did not show significant activation above the threshold for several days before the ensemble activity reemerged (Fig. 5c). The median of the ensemble lifetimes and the neuron lifetimes were 4 and 24 days, respectively ($n = 79$ ensembles, Fig. 5d). Thus, the ensemble lifetimes were significantly lower than the duration where more than two-thirds of the neurons in the ensemble were detectable (Wilcoxon rank-sum test $p = 1.41 \times 10^{-10}$), and the ensemble lifetime measurements were not limited due to our ability to track the individual neurons within the ensembles.

We examined the tuning of the identified ensemble patterns to SWRs. Some of the ensembles showed significant activations during SWRs (Fig. 5e). We further investigated the relationship between the strength of ensemble activation by SWRs and the ensemble lifetime during the three months (Fig. 5f). The activation strengths of short-lived and long-lived ensembles during SWR had a significant difference (Wilcoxon rank-sum test, $p = 0.0027$). The median activation strength for ensembles with shorter lifetimes (<18.9 days, mean + s.d.) was 26.5, while for longer-lived ensembles (>18.9 days, mean + s.d.), it was 0.8. Furthermore, the lifetimes of ensembles with low (activation strength <2.9, mean + s.d.) and high activation strengths (activation strength > 2.9, mean + s.d.) during SWRs were also significantly different (Wilcoxon rank-sum test, $p = 0.0045$). The median lifetime for the group with high and low activation strengths during SWR were 5.5 and 33.5 days, respectively. We found that 83.3% of the ensembles with higher activation strength during SWRs did not exhibit long lifetimes and consisted entirely of hippocampal ensembles. In contrast, neuronal ensembles that demonstrated significantly longer lifetimes (5 mixed assemblies, 3 purely HPC, and 1 purely PrL) were less tuned to SWRs and comprised neurons from both cortex and hippocampus (Fig. 5f).

## Discussion

We developed UFTEs, ultra-flexible electrode arrays with minimal footprint and capable of months-long stable recordings of high-SNR single unit activity from multiple superficial and deep brain areas in awake, freely moving rodents. UFTEs enabled long-term tracking of multi-areal neuronal ensembles with different lifetimes and tuning patterns to sharp-wave ripples.

Stiff electrode arrays can result in transient microglial activation[31], glial scar formation by astrocytes[32], and loss of neurons near the implant[6]. As a result, there is a decrease in the SNR and the yield of units recorded in the days after implantation of various stiff electrode arrays[7]. Likewise, micro motions in the brain on the order of several micrometers are caused by respiration and pulsation[33]. These can induce drifts in the amplitudes of spike waveforms recorded with stiff electrodes[12] and cause additional damage to the brain tissue near stiff implants[34]. Tungsten or platinum-iridium microwires in geometrical arrangements similar to our electrode arrays have yielded stable recordings and long-term tracking of neurons in some cases[35–37]. However, our results with UFTEs demonstrate superior SNR, yield, and unit-tracking capacity, most likely due to their ultra-flexibility and minimal implant footprint compared to the rigid metallic wires[38,39]. Additionally, the use of micro-fabrication techniques reduces the time and effort to construct UFTEs at mass scale, compared to microwire arrays that are assembled manually.

Recent ultra-high-density arrays, such as silicon polytrodes[11] or Neuropixels, spatially oversample the extracellular waveform of recorded neurons and, together with drift-tracking algorithms[12], can permit unit tracking despite drifts for up to 1–2 months. However, correcting for drifts through spatial oversampling comes with multiple shortcomings compared to minimizing the movement of the brain relative to the electrodes, as in UFTEs. First, the lack of motion between the tissue and the electrode obviates the need for spatial sampling and allows the number/density of recording contacts dedicated to each brain area to be freely chosen based on experimental goals. Furthermore, non-coaxial drifts of the probe with respect to the brain tissue cannot be overcome with oversampling and drift correction. Another shortcoming is that the chronic damage by the stiff substrate of the electrode array is likely to lead to a long-term decline in the numbers and SNR of single units recorded by rigid arrays, even with ultra-high-density oversampling[40].

It is possible to divide the current field of flexible electrode arrays into four general categories: (1) Single flexible shanks implanted by mechanical tethering to stiff shuttles[17,41], where incorporating many channels causes significant enlargement of the shanks and reduction in biocompatibility (the cross-sectional area of the shank scales linearly with the increasing numbers of recording contacts); (2) Multiple flexible shanks/fibers implanted by chemical tethering to stiff shuttles[14,15,42,43], which have significant limits on insertion depth because of the tradeoff between holding fibers together versus retracting insertion shuttles during time-sensitive surgical operations; (3) Flexible electrodes stiffened and implanted without stiff shuttles[44–46] which come with the significant risk of causing insertion damage due to the large glue footprint necessary for the sufficient stiffness to target deep brain regions. (4) Mesh electrodes[16,45,47], which have very large insertion footprints, where cross-section per recording contact is ~32.5 times larger for mesh electrodes[16] (600 μm width and 72% filled area for a 32-channel array) compared to the UFTEs. This limits the use of mesh electrodes in young brains to achieve long chronic recordings, where the developing brain can still recover from insertion. In contrast, the footprint of the implant scales with the square root of the number of recording contacts in UFTEs as the electrode fibers can be packed into a cylindrical bundle during insertion.

The main drawback of tethering flexible electrode arrays to stiff shuttles purely by chemical means is the delicate balance between the inherent limit to how deeply the arrays can be implanted in the brain (Supplementary Table 1) and the time until the insertion shuttle is decoupled from the arrays and can be removed from the brain during time-sensitive surgical operations[43,46,48]. We avoided operating within this narrow regime by decoupling the dynamics of electrode insertion and the retraction of the insertion shuttle. Since the shuttle retraction does not rely on the dissolution of the silk fibroin that is gluing the fibers in UFTE bundles, we can keep the UFTE bundle fibers together with strong glue/adhesion while we can still retract the insertion shuttle immediately after insertion.

Based on our experience, the triple coating of the electrode bundle with silk fibroin solution takes several hours to a day to dissolve, which is readily beyond the timescales of electrode insertion into any possible target area, even in large animal models or humans. Furthermore, our electrode insertion method can be realized with any other choice of biodegradable coating material that would keep the electrode fibers together for similarly long durations. On the other hand, the tungsten shuttles used in our current study, which had a 50-μm diameter, may not have sufficient stiffness to reach subcortical brain areas in large animal models or humans. In that case, these shuttles can be simply replaced by shuttles with a larger diameter or made of materials with significantly higher Young's modulus.

In addition to the high quality and stability of long-term recordings enabled by our electrode arrays, our surgical method and the

overall implant package provided significant advantages. Thanks to the flexible ribbon cables that connect each electrode bundle to the headstage, each bundle can be inserted into an arbitrary position of the brain at any angle once the electrode array is assembled. The TitaniumHelmets provide better protection to the electronics and the electrode arrays within and allow us to house the rats together (Supplementary Video 2), significantly reducing stress[49]. The custom Intan-based headstage allows direct soldering of the electrode arrays to the headstage, eliminating adapters between the electrode arrays and the amplifiers. Multiple such boards are stackable, enabling the user to distribute even larger numbers of channels among brain areas. The magnetically-guided recording cap can be easily plugged to the headstage and can be made entirely wireless, enabling near-continuous recordings in the home cage.

The exceptionally high SNR and yield of the single units recorded by the UFTEs can be attributed to: (1) the slow insertion of the electrode bundles into the brain (enabled by decoupling fiber-glue dissolution kinetics from shuttle retraction), which yield higher-quality recordings in vivo[50]; (2) the small footprint of UFTEs during insertion due to the self-assembly of the electrode fibers into a compact bundle by elastocapillary forces; (3) the ultra-flexibility of the individual UFTE fibers, which made them much less likely to induce immune reactions and push away neurons (Fig. 4f), likely resulting in cases such as the "cell-attached" recording (Fig. 3) where the soma of a neuron in possibly sealed the recording contact[51], which is also supported by the impedance magnitude of 7.88 MΩ at 1 kHz frequency that is significantly higher than the impedance magnitudes of other electrode sites where single-unit activity is typically recorded (300 kΩ–1 MΩ) or the gold electrode sites of equal size without any PEDOT:PSS coating (~2–3 MΩ); (4) PEDOT:PSS coating with additional steps to enhance cross-linking, charge mobility[52], and stability of attachment to the roughened gold recording contacts[53]; (5) inserting the electrode bundles through the dura mater with the combination of mechanical tethering and glue, which reduced damage to the brain surface and prevented edemas. (In our tests, ultra-flexible electrode fibers coupled to shuttles purely by biodegradable coatings were separated from the shuttle by the dura mater, no matter how strong the coating is).

To date, there have been few methods, yet no widely accepted standards for tracking single units across sessions from extracellular electrophysiology data[54,55], likely due to the rarity of such data. The recent emergence of ultra-flexible electrode arrays that enable long-term recording of single units necessitates robust methods for multi-session neuronal tracking. We found that the most reliable method towards this end is to perform the spike sorting on the entire dataset after concatenating sessions into one pseudo-session. We consider this method more robust than other methods based only on correlating spike waveforms across sessions to estimate the tracking of single units[16]. Nevertheless, we still performed additional quantitative tests to evaluate the inter-session similarity of tracked single-unit waveforms inspired by previous studies[54,55]. These tests verified that the spike waveforms of single units at different sessions had a significantly higher similarity than the spike waveforms of other single units. We did not consider the firing statistics since these can change across brain states or days[56,57] and, therefore, are unreliable parameters for tracking single units across sessions[54].

Even though a significant portion of the single units could be tracked for the entire duration of our experiments, we still observed some dynamicity in the detected single units across sessions. One common scenario was the disappearance of single units after one session or multiple sessions of tracking (which could reappear later) or the detection of new single units on a recording contact while the other single units on the same contact remained stable. Such disappearances or appearances of single units could be attributed to drastic changes in the firing rates of the recorded neurons[58], their movement[59], or death. In the cases where the entire composition of single units changed on a recording contact, the movement of the electrode fiber with respect to the surrounding brain tissue is possible. These phenomena can be investigated more rigorously by multimodal techniques where the positions and activity of the neurons surrounding the UFTEs can be imaged by optical methods parallel to the electrophysiological recordings. However, the UFTEs certainly provided near-constant stability in terms of the single-unit yield per recording contact and the quality of recorded single units throughout the 3.5 months.

Our findings indicate that some ensembles are less co-active with SWRs, are inter-areal, i.e., cortico-hippocampal, and long-lasting (up to three months). On the other hand, ensembles co-active with SWRs tended to be purely hippocampal with shorter lifetimes (a few days). Various studies show that CA1 pyramidal neurons exhibit two highly distinct firing patterns with distinct neurochemical profiling, which correlate with their location in either the superficial (CB positive sublayer) or deep (away from the CB positive sublayer) layers of the CA1 pyramidal layer[60,61]. Mizuseki et al. observed that pyramidal cells located in deeper layers fire at higher rates, exhibit more frequent bursting, and are more significantly modulated by slow-wave sleep oscillations than their superficial counterparts[62]. Following this, Grosmark and Buzsaki identified a distinction between fast-firing pyramidal neurons, which form a more rigid group in terms of plasticity with low spatial specificity, in contrast to slow-firing neurons, which are more plastic, gain high spatial specificity during exploration, and demonstrate a stronger association with sharp-wave ripples (SWRs)[63]. Danielson et al. found that superficial hippocampal place maps were more stable during exploration, while deep hippocampal place maps are preferentially stabilized during goal-oriented learning tasks[64]. Studies from Soltesz's lab emphasize the specialized role of deep CA1 pyramidal neurons, which are more involved in processing environmental landmarks, stabilizing goal-directed place maps, and tracking reward configurations[65]. Also, they describe that the mPFC-projecting pyramidal cells were almost exclusively located in the deep sublayer[20]. Fernandez-Ruiz and colleagues concluded similarly that deep temporal CA1 pyramidal cells project directly to the mPFC, and they described that superficial cells primarily contribute to the stable representation of spatial contexts and the encoding of future choices[66,67]. Gava et al. showed that deep high-activity cells established co-firing motifs that remain stable across subsequent experiences[68]. The CA1 neurons we observe with long ensemble lifetimes, strong coupling to mPFC, and weak tuning to SWRs could be the less plastic, high-firing, deep CA1 pyramidal neurons with projections to mPFC. Indeed, hippocampal ensembles with such long lifetime are also likely functionally important: Goshen et al. demonstrated that real-time inhibition of the CA1 region can impair month-long remote memory recall while sparing recent memories[69], whereas Atucha et al. found that optogenetic inhibition of the CA1 region can impair remote memory recall 6–12 months after memory formation[70]. The existence of such CA1-cortical interareal ensembles with months-long lifetimes challenges the view that exclusively cortical ensembles maintain long-term memories.

UFTEs can be used to advance our understanding of the neuronal dynamics underlying learning and long-term memory in distributed brain networks. UFTEs can also be used in the clinic to help minimize tissue damage, eliminate immune response, and enable lifelong implants and stable high-performance decoders for brain-machine interfaces. High-resolution, stable recordings of single-unit activity provided by UFTEs can improve our understanding of brain disorders, enable the development of new diagnostics, and lead to the discovery of efficacious treatments that would otherwise be combinatorially impossible to discover using behavioral readouts alone. We recently developed a version of the UFTEs to record 3 cm deep in the human brain, which will be used for investigating the physiology of epileptogenic tissue before surgical removal.

## Methods

All experimental and surgical procedures involving animals were approved by the local veterinary authorities of Canton Zurich, Switzerland, and were carried out in accordance with the guidelines published in the European Communities Council Directives 2010/63/EU.

### Fabrication and characterization of the UFTEs

Supplementary Fig. 1 shows the device fabrication steps. We spun PI2610 (HD Microsystems) on a 4-inch silicon wafer to form a 1.2-μm-thick polyimide layer. After curing the polyimide film in a programmable oven (CLO-2AH-S, Koyo ThermoSystems) at 300 °C, we patterned ma-n-1420 (Micro Resist Technology GmbH) on this polyimide film by direct laser lithography (Heidelberg Instruments DWL 66 + ). The solder pads, electrode contacts, and wires were patterned by lift-off after depositing 10 nm titanium and 150 nm gold with electron-beam evaporation (Plassys MEB550S). We coated the metal layer with a 1.2 μm-thick polyimide insulation layer and patterned AZP4620 (MicroChemicals GmbH) on it using direct laser lithography to create the canals between electrode fibers, device borders, and solder pad openings. We etched the polyimide in exposed areas in $O_2/CF_4$ plasma (Plasmalab 80 Plus, Oxford Instruments) and removed the excess photoresist.

To pattern PEDOT:PSS coating on the electrode contacts via a dry lift-off process, we coated a 2.5 μm layer of sacrificial parylene C on the wafer by chemical vapor deposition (PDS 2010 Labcoter 2; Specialty Coating Systems Inc.). We spun a layer of 2% Micro-90 solution (International Products Corporation) on the wafer before parylene coating to facilitate the separation of the sacrificial parylene later. Afterwards, we patterned AZP4620 on the parylene C by direct laser lithography, coating everything except the recording contacts. Then, we etched the parylene C and polyimide on the recording contacts in $O_2/CF_4$ plasma. We placed the wafer briefly inside a diluted gold etchant ($KI/I_2$, FIRST Micro- and Nanotechnology Center, ETH Zurich) to roughen the gold on the recording contacts and enhance the adhesion of PEDOT:PSS to the electrode contacts[53]. We rinsed the wafer in ultrapure water and removed the excess photoresist. We spun a mixture of PEDOT:PSS (Clevios PH 1000, Heraeus Epurio), ethylene glycol (Sigma Aldrich), dodecyl benzene sulfonic acid (Sigma Aldrich), and (3-Glycidyloxypropyl) trimethoxysilane (Sigma Aldrich) on the wafer at 650 RPM, and placed the wafer in an oven at 140 °C for one hour[52]. We repeated the spinning/baking of the PEDOT:PSS mixture two more times to achieve a total thickness of 450 nm. After the wafer cooled down, we peeled off the parylene C, leaving PEDOT:PSS only on recording contacts.

We characterized the electrodes by optical microscopy (Nikon Eclipse L200D) and surface profilometry (DektakXT, Bruker Corporation) at various process steps. We measured the impedances of the electrodes in saline and in vivo with the impedance measurement function of the electrophysiological recording system (Intan Technologies). According to the datasheets of the RHD2164 chip and RHX Acquisition Software, and the source code provided by Intan Technologies, a sinusoidal current wave of desired test frequency was generated by coupling a digital-to-analog converter (DAC) to a capacitor (0.1 pF, 1 pF, or 10 pF) and injected into the electrode. The resulting voltage across the electrode and the saline/tissue was measured by the corresponding amplifier input channel during the impedance measurement.

All impedance measurements were performed in a two-electrode setup. To generate the impedance spectroscopy shown in Fig. 2c, we inserted the electrode fibers in ringer solution (B. Braun), and ran impedance measurements at frequencies ranging from 2 Hz to 5 kHz. The reference/counter electrode was an Ag/AgCl wire also immersed in the saline along the electrode arrays. For the in vivo impedance measurements shown in Fig. 4c, the reference/counter electrode was the 0.9 mm-diameter stainless steel screw placed on the cerebellum

(approximate coordinates: −12.5 mm AP, 2.5 mm ML), which also served as the reference during the recordings.

### Assembly of the UFTEs

After peeling the UFTEs off the silicon wafer, we aligned their solder pads with those on the headstage and soldered them together at 270 °C (Supplementary Fig. 2a). After verifying the soldering quality by impedance measurement in saline, we dipped the electrode bundles in 0.2 g/ml PEG4000 (Sigma Aldrich) in double distilled water. We painted each electrode bundle with silk fibroin solution (50 mg/ml aqueous solution, Sigma Aldrich) three times while avoiding clogging the loops at the tips (Supplementary Fig. 2b).

We prepared tungsten shuttles by cutting 15 mm segments from straight-cut tungsten wires of 50 μm diameter (W5606, Advent Research Materials). We partially inserted the tungsten wire segments into glass capillaries with pulled tips for easier handling. After trimming the tungsten wires to the desired length, we thinned a 500 μm-long portion of the tungsten wire at the tip to 20 μm diameter by electrochemical etching in 0.9 M KOH. Afterwards, we sharpened the tip of the tungsten wire by applying a 2 V DC to the wire while a 250 μm portion of its tip was in the KOH solution. Finally, we rinsed the tips of the shuttles in deionized water and verified the tip dimensions and sharpness under a microscope.

After soldering the UFTEs to the headstage (Supplementary Fig. 2a) and preparing the tungsten shuttles, we attached the headstage to a 3D-printed electrode holder, which was attached to a stereotaxic arm. We also attached one of the tungsten shuttles to the opposite stereotaxic arm via a 3D-printed pipette holder. Afterwards, we inserted the tip of the shuttle inside the loop of the most anterior electrode bundle by stereotaxic maneuvering (Supplementary Fig. 2b). For additional mechanical stability, we fixed the ribbon cable of that bundle to the body of the glass pipette of the shuttle with the 0.2 g/mL PEG4000 solution. Then, we transferred the shuttle from its stereotaxic arm into its respective slot in the electrode holder (Supplementary Fig. 2c). We repeated the process for the other UFTE bundles.

### TitaniumHelmet and surgical procedures

Headley et al. developed a cap for rats with a total weight of 16.9 g made from Acrylonitrile butadiene styrene (ABS)/polylactic acid (PLA) plastic, which is not biocompatible. According to the author, this biocompatibility problem was mitigated because dental acrylic was placed between the 3D-printed components and the skull[71]. There is also a lighter rat cap developed by Vöröslakos et al., which is only 8.3 g and made from clear v4 (RS-F2-GPCL-04, Formlabs) resin which is also not biocompatible[72]. This cap has a smaller area for craniotomies with a fragile wall and requires unscrewing to open the cap. However, when aiming for long-term recordings lasting more than 2–3 weeks and continuous recordings, factors such as full biocompatibility, protection from other animals in the home cage, quick assembly/disassembly, accessibility to the electronics, and reusability need to be all considered. In our design, we applied the same principles essential for primates[73]. The TitaniumHelmet has four parts: the base (0.25 g), left-right (5 g, 8 g) enclosures, and the top cover (8 g) (Supplementary Fig. 3b), each made of grade 5 Titanium (Ti-6Al-4V) commonly used in medical applications. Only the base is cemented to the skull, while all the other parts can be disassembled and reused for further experiments. The total mass during recording without the cover part is 13.7 g. The maximum anterior-posterior extent of the base is 23.95 mm, and the maximum lateral extent is 11.30 mm between the left and right temporal crests (Supplementary Fig. 3a). The base has two front screw holes (1 mm Ø) and one rear screw hole (1 mm Ø), compatible with titanium screws (0.9 mm Ø, length 3 mm - M-5100.03 Medartis) for skull attachment. On the lateral side of the base, there are five holes with threads (M1.2). The left and right shells attach to the base with screws. The 256-channel custom headstage PCB (0.45 g) fits

into the inner edge rails of the shells and remains in place as part of the TitaniumHelmet throughout the rats' lifetime. In addition, the shells are held together by front and rear screws. The front of the inter-connected enclosures is designed to hold the cover in place with an additional rear magnet. There are two top covers: a titanium one for protecting the headstage when the animals are in the home cage, and another 3D-printed one (RS-F2-GPCL-04 clear resin, Formlabs) con-taining a PCB that can be connected to the headstage and transmits the digital signals to the custom-made FPGA board during the recording (Supplementary Fig. 3d).

The rats that were used in this study were female Long Evans rats ($n = 4$ rats, 21-59 weeks of age, 270-340 g of weight at the time of surgery, Janvier Labs and Charles River Laboratories). Female rats were used in this study due to the lower risk of fighting with cagemates. The rats were housed in groups in standard IVC cages (Allentown), and had ad libitum access to food and water. They were kept on an inverted light cycle (12 h dark/12 h light) at a temperature of 23 °C. The humidity in the room and in the cage were 52% and 58%. We anesthetized the rat with isoflurane (Attane, Piramal Pharma Ltd.) mixed in oxygen. Meloxicam (Metacam, Boehringer Ingelheim) was injected sub-cutaneously as an analgesic. Bupivacaine (Bupivacain Sintetica, Sinte-tica) was subcutaneously injected in the scalp as a local analgesic. A mixture of Ringer's solution and glucose (Aequifusine, B. Braun) was also injected subcutaneously on a regular basis during the surgery. We shaved the head of the rat and cleaned its scalp with Betadine (Mun-dipharma Deutschland GmbH). After fixing the rat's head in the ste-reotaxic frame, we incised the skin and cleared the connective tissue to expose a sufficiently large area on the skull. We ensured the parallelity of the skull with the ground and identified the locations of the cra-niotomy holes. We drilled these holes (three holes for 0.9 mm screws holding the base of the TitaniumHelmet, three holes for 1.5 mm screws anchoring the base to the skull, two holes for ground and reference screws, and four holes for electrode implantation sites). We secured the base to the skull using titanium screws and dental cement. After-wards, we implanted the UFTE bundles as described in the Results (Supplementary Fig. 2d-e). We used a predefined implantation sequence from the anterior to posterior bundles, with the most ante-rior bundle being the first to be implanted. If two bundles were implanted next to each other in the same coronal plane of the brain, the more medial one precedes the more lateral one. Once we implanted all four UFTE bundles, we covered the electrode implant sites with a silicon elastomer (KwikCast, World Precision Instruments) and transferred the headstage to the TitaniumHelmet. We closed the TitaniumHelmet, cleaned the wound, and sutured any gaps in the scalp. We stopped the isoflurane anesthesia and let the rat wake up in a clean, warm cage with wet food pellets, bedding, and nesting material.

In the case of multiple headstage boards, one can implant bundles in the sequence described above, starting with the first headstage board (bottom of the stack). When the bundles connected to the first board are implanted, the headstage from the stereotaxic arm can be released (with the same male connector on the bottom side of each headstage) and temporarily held on the side with a holder. The second headstage, loaded with another group of UFTE bundles, can then be attached to the stereotaxic arm. The new bundles can be implanted either in the same hemisphere as the previous bundles by following the same anterior-posterior sequence as before or in the other hemi-sphere. Afterwards, the second headstage board can then be plugged into the first one by sliding the uppermost top headstage board into its respective slot inside the TitaniumHelmet. The shell of Tita-niumHelmet only holds the top headstage because this is exposed to make a connection with the recording hardware.

## In vivo recordings from rats
The rats were familiar with the environment (A 50x50x50cm plexiglas cage covered by copper mesh on the sides and bottom but opened on

the top side). The cage's floor was covered with bedding and changed after each recording session, which was conducted twice a week. We began the recording within a maximum of six days after surgery. Within the cage, a glass petri dish with a single drop of concentrated milk (Kondensmilch, Coop Switzerland) was a positive reward after connecting the recording system to the rat.

We designed a headstage that can be easily encapsulated by the TitaniumHelmet (Supplementary Fig. 2a). Our custom headstage enables the soldering of UFTEs to its bottom side (Supplementary Fig. 2b), handling a minimum of 256 channels. Multiple headstages can be stacked to record up to 1024 channels. The top side of the head-stage is equipped with four Intan electrophysiology integrated circuits (4xRHD2164 = 256 channels/head stage, Supplementary Fig. 2a). To communicate with the headstages, we assembled a host recording system based on a custom-developed board holding an Opal Kelly XEM6310 module (based on Xilinx Spartan 6 field-programmable gate array), providing identical functionality to the RHD-Series Amplifier Evaluation System. Between the headstages and the recording system, we used a small PCB with a connector for digital signals only, plugged into the encapsulated head-stage after opening the magnet-held cover of the TitaniumHelmet. The module ran Rhythm firmware from Intan Technologies (http://intantech.com/downloads). We used the RHX Data Acquisition Software to record broadband data at a 20,000 Hz/channel sampling rate at 16-bit resolution. A high-pass filter with 0.1 Hz cut-off frequency was applied at the hardware level to eliminate the DC component of the signal. In the last recording sessions for all rats, we also successfully tested a 512-channel wireless logger (data saved onto an SD card but not transmitted wirelessly) for up to one hour con-nected to the 256 channels in the implanted animals (Supplemen-tary Fig. 3c).

## Mouse implantation and recordings
Chronic electrophysiological recordings were performed in two Thy1-GCaMP6 male mice (2–3 months old, 25-30 g weight at the time of the surgery, Jackson Laboratory). Sex was not considered in the study design since the goal was testing UFTEs. The mice were kept in a reversed dark/light cycle (12 h light/12 h dark) at a temperature of 22 °C. The humidity in the room and in the cage were 50% and 59%. Implantation targeted the CA1 subfield of the dorsal hippocampus. During implantation, animals were anesthetized with isoflurane (2–3% for induction, 1–2% during surgery, Piramal Pharma Ltd.), sub-cutaneously injected with medetomidine (Domitor Orion Pharma) as an analgesic, and their body temperature was maintained using a heating pad (DC Temperature Controller 40-90-8D, FHC). Topical lidocaine (Emla Creme, AstraZeneca) was applied to the skin for local anesthesia. The scalp was retracted, and the skull was exposed and sealed with dental acrylic. A small craniotomy was performed over the cerebrum (AP: −3.6 mm, ML: 3.2 mm), and the probe was inserted into the brain by tethering to either a 100 μm fiber optic cannula or a 50 μm tungsten insertion needle. Two additional trepanations were per-formed over the cerebellum, and silver wires were placed in contact with the CSF to serve as ground and reference electrodes. After implantation, the probe was fixed with additional acrylic, and the connector was affixed to the animal's head. The animal was allowed to recover for one day after the surgery, and then recording proceeded regularly for the duration of the experiment. The animal was head-fixed and placed in an enclosed, soundproof box during recordings. For electrophysiological recording, the voltage was amplified and digitally sampled at a rate of 30 kHz using a commercial extracellular recording system (TDT digital ZIF-clip headstage, Tucker-Davis Tech-nologies and RHD2000 Recording System, Intan Technologies).

## Immunohistological processing of the brain tissue
At the end of chronic experiments, we euthanized the rat with an intraperitoneal injection of 300 mg/kg sodium pentobarbital

(Esconarkon, Streuli Tiergesundheit AG). Once the rat was under deep anesthesia, we performed transcardial perfusion with 4% paraformaldehyde solution in phosphate-buffered saline (PBS). After extraction from the skull, we stored the brains inside a 4% paraformaldehyde solution for post-fix. We sliced the brains into 100 μm-thick slices with a vibratome (Leica VT1200S, Leica Biosystems). After washing the slices with PBS three times, we placed the slices into a primary antibody mixture of Rabbit-anti-IBA1 (1:1000 dilution, 019-19741, FUJIFILM Wako Pure Chemical Corporation) and goat-anti-GFAP (1:500 dilution, ab53554, Abcam) in a blocking buffer. We incubated the slices in this primary antibody mixture at 4 °C temperature for one week. We washed the slices in PBS three times and placed them into a secondary antibody mixture comprising goat-anti-rabbit Alexa Fluor Plus 488 nm (1:1000 dilution, A32731, Thermo Fisher Scientific) and Neurotrace 640/660 nm (1:500 dilution, N21483, Thermo Fisher Scientific) in a blocking buffer. After three days of incubation in this secondary antibody mixture at 4 °C temperature, we washed the slices in PBS three times. We placed them into another secondary antibody mixture of donkey-anti-goat Alexa Fluor 405 nm (1:1000 dilution, ab175664, Abcam) in a blocking buffer. After three days of incubation in this secondary antibody mixture at 4 °C temperature, we washed the slices in PBS three times and mounted them on glass microscope slides. We used iohexol (350 mg/ml) as the mounting medium. In all stainings, the blocking buffer consisted of 1% bovine serum albumin and 0.1% Triton-X−100 (Sigma Life Science) in PBS.

Serial 100 μm-thick brain slices were imaged with a confocal spinning disk microscope (IXplore Spin 50 μm, Olympus) with a z-step size of 5 μm using a 20X 0.8 NA air objective lens (UPLXAPO20X, Olympus). We acquired the images with cellSens Dimension (version 2) software from Olympus. We used 405 nm (50 mW), 488 nm (60 mW), 561 nm (60 mW), and 640 nm (60 mW) laser lines (OBIS, Coherent) for fluorescence excitation. We captured the images using a CMOS camera (Prime BSI Scientific sCMOS) with 2048 x 2048 pixels as 16-bit images. We stitched the single image tiles into a mosaic image of the whole rat brain slice. We combined the images from each fluorescence channel to form a multichannel composite.vsi image using cellSens Dimension software (Olympus). Subsequently, we imported the composite.vsi files to ImageJ[74] and converted them to .tiff files using the Bio-Formats plugin[75].

For quantifying the chronic effects of the electrode arrays on the brain tissue, we first generated an average intensity projection of the image slices in the brain slice that contained an electrode bundle. We then binned the image 4x4. In the resulting image, we defined regions of interest (ROI) with 25 μm radial steps from the location of the electrode bundle in the brain slice. Then, we randomly selected 1000 sample pixels among the pixels in the ROI and calculated the mean and s.e.m. (standard error of the mean) of the fluorescence intensity of these pixels in each of the ROIs. Finally, we normalized these mean intensity values by dividing them by the mean fluorescence intensity value of the ROI containing points with 500 μm distance from the electrode bundle, which we used as a control. We repeated this procedure for all fluorescence channels.

### Analysis of the electrophysiology data for single-unit sorting and tracking

We used JRCLUST 4.0.0[28] for spike-sorting on selected recording sessions. First, the spike sorting pipeline filtered the raw data with a 4th-order bandpass filter with the cutoff frequencies at 300 and 5000 Hz. Then, it performed a common average referencing on the filtered data by computing the median across the traces of all intact channels and subtracting this median from the filtered trace of each intact channel to eliminate the artifacts from instrumentation or the strong muscle movements of the rat. Afterwards, it detected spikes on the filtered and software-referenced traces as described by Quian-Quiroga et al.[76] (qqFactor=5, only negative peaks detected). Events detected within a 60 μm spatial and 0.25 ms temporal vicinity ("evtDetectRad,"

"evtMergeRad," and "refracInt") were merged into one spiking event to prevent the detection of duplicate spike events from multiple recording contacts. We reduced the dimensionality of the detected spike waveforms by principal component analysis (3 features per recording contact, "nPCsPerSite"). Spikes were clustered automatically by using the Density Peak clustering algorithm[77], where logarithms of rho and delta cutoffs were −2.5 and 0.6, respectively ("log10RhoCut" and "log10DeltaCut"). After the automatic clustering was complete, we performed a manual curation to eliminate noise clusters and finalize the cluster identities of spikes.

To robustly test the quality of the sorted single units, we used three parameters: SNR, percentage of ISI violations, and distance from the nearest neighbor. We calculated the SNR by dividing the absolute value of the amplitude of the mean spike waveform of each unit (at the recording contact where the unit has the highest amplitude) by the root-mean-square of the bandpass-filtered signal at the corresponding recording contact. We calculated the percentage of ISI violations by calculating the time intervals between all consecutive spikes in each unit (also known as the interspike interval), counting the instances where these time intervals are less than 2 ms, and dividing the number of these instances by the total number of interspike intervals. To calculate the distance from the nearest neighbor, we first iterated over recording contacts to perform the principal component analysis on the spike waveforms on that recording contact. We reduced the dimensionality of each spiking event from (number of recording contacts) x (40 samples) to (number of recording contacts) x 3. Afterwards, we calculated the Mahalanobis distances between the centers of each unit cluster and other clusters in the recording contacts in the neighborhood of the recording center, that is the center of the unit cluster ±3 recording contact. We identified the cluster with the smallest distance to the cluster of interest and recorded this distance as the "distance from the nearest neighbor."

We calculated the Pearson's correlation coefficient between two mean waveforms $i$ and $j$ as follows:

$$R_{ij} = \frac{C_{ij}}{\sqrt{C_{ii}C_{jj}}} \qquad (1)$$

where $C_{ij}$ is the covariance between $i$ and $j$, $C_{ii}$ and $C_{jj}$ are the variances of $i$ and $j$ respectively. We calculated the standardized mean difference between two clusters $k$ and $l$ as the following:

$$SMD_{kl} = \sqrt{\sum_{d=1}^{D}\left(\frac{m_{k,d} - m_{l,d}}{2\sqrt{\sigma_{k,d} + \sigma_{l,d}}}\right)^2} \qquad (2)$$

where $m_{k,d}$ and $m_{l,d}$ are the means across spikes of clusters $k$ and $l$ in the $d^{th}$ principal component axis, $\sigma_{k,d}$ and $\sigma_{l,d}$ are the variances across the spikes of clusters $k$ and $l$ in the $d^{th}$ principal component axis, and $D$ is the total number of principal components (3 x 64 recording contacts per electrode bundle).

### Sharp-wave ripple detection and alignment

Eight LFP channels were selected, starting from stratum radiatum in CA1, where sharp-waves (the large amplitude negative polarity deflections with 40−100 ms duration) were recognizable, followed by channels where ripples could be detected in the CA1 pyramidal layer, and channels from stratum oriens where the positive deflection of a sharp-wave component was observable. We first downsampled the data to a sampling rate of 2 kHz to analyze the oscillations in the local field potential oscillations. We automatically detected SWRs using a script (bz_FindRipples.m, publicly accessible on GitHub (https://github.com/buzsakilab/buzcode) initially developed by Hajime Hirase and Michaël Zugaro (https://github.com/michael-

zugaro/FMAToolbox). This script identifies ripples by applying the normalized squared signal (NSS) technique, which entails thresholding the baseline, merging nearby events, thresholding the peaks, and discarding events with excessive duration[78–80]. We also cross-validated our results with the recently developed sharp wave-ripple detection algorithm that adapted a CNN architecture to search for SWR in the hippocampus[81]. For manual curation of automatically detected SWRs, an interactive graphical user interface was developed using MATLAB's figure-based framework, App Designer. This tool allows straightforward browsing through multichannel LFP traces and other derived signals, such as the bandpass-filtered LFP, power ripple components, and wavelet transformation of given channels from different layers of the hippocampus. SWRs can be manually inspected or annotated by the event start and end point specification.

SWR events can be defined as a series of intervals, with each SWR characterized by its onset and offset points. However, identifying the precise borders of each SWR can be challenging. We detected the SWR intervals and aligned them to a central time point. To accomplish this, we utilized a two-step procedure: First, we identified the peak of the bandpass signal power, which was averaged over all SWR channels. In the second step, we aligned a fixed-length window of the mean bandpass LFP around the power peak to a template, using maximization of cross correlation. We limited signal shifts to a maximum of 10 frames (or 5 ms at 2 kHz) to prevent shifts greater than one period of a 200 Hz signal. Aligned SWR episodes were used to calculate the frequency decomposition using wavelet transformation for further analysis.

### Identification of neuron ensembles

An unsupervised statistical framework based on independent component analysis was used to detect patterns of co-firing between neurons in different cortical and hippocampal areas. Spikes from each recorded neuron were counted in 25 ms time bins, and then the spike counts were z-scored ($Z$), $Z_{i,j}$ representing the activity of neuron $i$ during time bin $j$. Principal components were computed by eigenvalue decomposition of the correlation matrix $C = \frac{ZZ^T}{N}$ of $Z$, where $N$ is the number of time bins (25 ms) of $Z$. To extract ensemble patterns, a two-step procedure was followed. Initially, the number of significant cell ensembles (which refer to a subset of neurons with correlated activity) was estimated by computing the eigenvalues of the principal components of the correlation matrix ($C = \sum_{i=1} \lambda_i x_i x_i^T$ where $x_i$ is the $i$-th eigenvector of $C$, in other words the $i$-th PC of $Z$, and $\lambda_i$ its corresponding eigenvalue) that exceeded the Marčenko-Pastur threshold derived from an analytical probability function[30]. Subsequently, an independent component analysis was employed to extract the ensemble patterns by projecting the data onto the subspace spanned by the significant principal components and then computing the independent components through the fastICA algorithm[30,82,83].

We identified members of the cell ensembles using Otsu's method by dividing the absolute independent component (IC) weight into two major groups that aimed to maximize inter-class variance[84]. Neurons belonging to the group with a higher absolute weight were then classified as members of the neuronal ensembles.

To investigate the cortical responses of ensembles during SWRs, we computed the instantaneous ensemble activation strength as:

$$A_i(t) = z_i(t)^T . f\left(W_i^T . W_i\right) . z_i(t) \tag{3}$$

where $W_i$ represents the weights of members belonging to the $i^{th}$ ensemble and $z_i(t)$ refers to the activity of the ensemble members at each time $t$ (25 ms bin). Additionally, $f(W_i^T . W_i)$ represents a transformation of the outer product, with the diagonal set to 0 to avoid high activation strengths resulting from spiking by a single neuron. Ensembles were considered active when their activation strength exceeded a threshold corresponding to the 2 x s.d. of values above the baseline.

### Statistics and reproducibility

Student's $t$ test and Wilcoxon's Rank-Sum test (two-tailed) were performed to analyze electrophysiological data. Normality was tested with the Kolmogorov-Smirnov test. If the distribution was normal, the Student's $t$ test was done. Otherwise, Wilcoxon's Rank-Sum test was performed. Kolmogorov-Smirnov test was performed to compare pixel intensities in the imaging data. The statistical analysis was performed using MATLAB 2023a (MathWorks, Natick, MA, USA) and the SciPy package for Python 3 (https://scipy.org). No statistical method was used to predetermine sample size. The impedances of broken channels in Fig. 2b, Fig. 4c, Supplementary Fig. 7 and Supplementary Fig. 9a-b were excluded from the impedance statistics (exclusion criteria provided in Results and corresponding legends). The spike clusters that were not identified as single units during the spike sorting process and did not pass the ISI violation criteria were classified as multi-units and were excluded from the ensemble and single-unit stability/quality characterization, as clearly stated in the Methods. The experiments were not randomized. The spike sorting process and the contributing author in charge of the manual curation of the sorting were blinded to the potential neuronal ensemble memberships of the single units. The ensemble analysis was performed by a different contributing author than who performed the spike sorting process.

Figure 1d demonstrates the two cases where we managed to capture parts of UFTE bundles intact in a brain slice after transcardial perfusion, removal of the brain, and tissue slicing/processing. That panel is for qualitative demonstration only. Immunohistochemical processing of brain slices is performed for 11 UFTE bundles in 3 rats, yielding similar qualitative results to Fig. 4g (see Supplementary Fig. 6 for examples). The quantitative analysis was done for the slice shown in Fig. 4g, due to the homogeneity of cell density around the UFTE bundle in mPFC compared to the ones in other structures.

### Reporting summary

Further information on research design is available in the Nature Portfolio Reporting Summary linked to this article.

## Data availability

All data supporting the findings of this study are available within the article and its supplementary files. Additional data is deposited to the Zenodo repository at: https://zenodo.org/records/11236154 (https://doi.org/10.5281/zenodo.11236153)[85]. Any additional requests for information can be directed to, and will be fulfilled by, the corresponding author. Source data are provided with this paper.

## Code availability

All custom code and some preprocessed data used in this manuscript are uploaded to the GitHub repository at: https://github.com/Neurotechnology-at-ETH-Zurich/UFTE_paper (https://doi.org/10.5281/zenodo.11246326)[86].

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

## Acknowledgements

This project has been funded by the Swiss Federal Institute of Technology (ETH) Zurich (internal funds, M.F.Y.), the European Research Council (ERC) under the European Union's Horizon 2020 research and innovation program (grant agreement No 818179, M.F.Y.), and the Swiss National Science Foundation (SNSF, Sinergia Project CRSII5_198739/1, M.F.Y.). Longitudinal mouse recordings were funded by the University of Zurich (Forschungskredit: FK19-041, C.M.L.). We thank Reto Maier and the Physics Workshop of the University of Zurich for their advice on designing and manufacturing the TitaniumHelmet. We thank the staff of FIRST Micro- and Nanotechnology Center, ETH Zurich, for supporting the microfabrication of the UFTEs. We thank the Laboratory Animals Service Center (University of Zurich) staff for their support in caring for the experimental animals. We thank Gyorgy Buzsaki, Loren Frank, Adam Kepecs, Flavio Donato, Nick Melosh, Antonio Fernandez-Ruiz, and Liset de la Prida for helpful discussions, and Eminhan Ozil and Mostafa Ghannad Rezaie for helpful comments. Special thanks to Natalia Magnusson for artistic representation of the electrodes.

## Author contributions

T.B.Y. and P.G. contributed equally to this project and have the right to list themselves first in bibliographic documents. T.B.Y., P.G., W.B., and M.F.Y. devised the electrode geometry and insertion method. T.B.Y. designed and microfabricated the UFTEs. T.B.Y. and P.G. assembled the UFTEs before the implant surgeries. B.W. and T.B.Y. performed initial

electrode insertion tests. P.G. designed the Titanium Helmet. P.G. and A.L.V. designed the custom headstage and the electrophysiological recording system. T.B.Y., P.G., and W.B. performed the electrode implant surgeries on rats. T.B.Y. and P.G. collected electrophysiology data from the freely moving rats. C.M.L. implanted the UFTEs in mice and collected the electrophysiology data from mice. F.H. supported the collection of electrophysiological data from mice. A.D.V. and T.B.Y. performed the immunohistology on the brain tissue after transcardial perfusion. A.D.V. imaged the immunostained brain slices. T.B.Y. pre-processed and performed spike-sorting on the electrophysiology data from mice and rats and analyzed the imaging data from immunohistology. P.G., L.M., and V.L. developed the pipeline for detecting and curating SWRs in the electrophysiology data. P.G. and T.B.Y. analyzed the electrophysiology data to investigate unit stability and ensemble detection/activation. T.B.Y., P.G., W.B., and M.F.Y. wrote the manuscript with input from all the co-authors. M.F.Y. is the corresponding senior author and the principal investigator of the study.

## Competing interests
T.B.Y., P.G., W.B., and M.F.Y. are listed as inventors in a patent application filed on 21 May 2024 by ETH Zurich that covers the assembly and insertion method of UFTEs. The remaining authors declare no competing interests.
