## [Peer Review File · Nature Communications]

REVIEWER COMMENTS

Reviewer #1 (Remarks to the Author):

Yasar et al. describe a novel electrode design for extracellular electrophysiological recordings consisting of ultraflexible electrode bundles and demonstrate remarkably stable recordings in rats over multiple months using this design. The authors are to be congratulated for a very impressive proof-of-concept manuscript on a tool that could be useful for a wide range of neuroscience researchers. The paper is well written and without major methodological flaws. The results regarding the long-term stability of assemblies are potentially interesting, although some caution is warranted given that they are based on two animals only.

I have two major concerns regarding the statistical analyses and the description of the methods used. Neither of these would alter the main conclusions of the paper, but in my opinion they should be remedied before the paper can be accepted. There are also some minor glitches and ambiguities that could be improved in a revised version, as outlined below.

Major

1. The description of the algorithms used for spike sorting are conflicting between the method's section and the main text. The text references Jun et al., Biorxiv and correctly describes the process underlying Kilosort, though it is unclear which version of the algorithm was used. The methods refer to JRCLUST and Tank et al. as a reference, which seems to be based on multi-unit activity. Please clarify which algorithm and version was used in the methods and cite the corresponding paper.

2. The statistical analysis used in Fig 4c (Wilcoxon Rank sum test based on arbitrarily chosen time points) is inappropriate in my opinion. First, the data points represent repeated measures of impedances in the same channels and paired testing would be indicated. Second, if individual time points are picked for comparison, some justification as to how they were chosen should be given and they should be consistent between the two animals (e.g. week 1 vs. week 4, week 4 vs. week 12 or similar). The correct form of statistical test to show that impedances do not change would involve some form of equivalency testing for time series.

On the other hand, I don't think that statistical testing is strictly necessary to begin with as the figure merely describes the stability of impedances over time and the relevant information is clear without p-values.

Minor

1. Line 63, Introduction: The authors should cite Schoonover et al., 2020, Nature, PMID 34108681 as a counterexample that used conventional silicon probes over multiple weeks with highly optimized surgical techniques. They may want to discuss this reference at some point later, too.

2. Line 189: The abbreviations IL, PrL and Cg1 should be introduced here (or somewhere else) as they are later used in multiple figure legends without ever being introduced.
3. Line 200: it is not entirely clear here or in the method what the earliest time point after surgery was at which the authors attempted to record. Figure 4c indicates that impedances may have been very low shortly after implantation. Therefore, this information would be interesting and should be added either here or in the methods.
4. Line 709: states the craniotomy for inserting the probe was made over the cerebellum but the text states that recordings in mice were made in CA1. Please clarify / correct as appropriate.
5. Fig 2a (v): Would recommend a higher magnification and higher resolution for this photograph, otherwise its hard to take anything away from it.

Reviewer #2 (Remarks to the Author):

Yasar et al. report on the development of a 256-channel flexible recording array with 4 bundles of physical separated electrodes capable of recording over multiple months. The channel count is impressive, in particular for a probe that makes good use of the material flexibility to avoid adverse reaction of the tissue to the implant. Additionally, the ability to probe 4 different brain regions is interesting and the developed method of implantation to enable choice of brain regions and implant angles is valuable. The paper is well written, the technology and data are presented well, and the demonstrated capability to record over months is significant. I have some questions/comments, but overall, I find that the manuscript is already in a good state.

Specific comments in the text/figures:

Page 3, Line 100 – does the comparison stating the mesh electrode has 22 larger cross-sectional area per recording contact include the “empty space” of the mesh? If so, is this a fair comparison?

What is the reason for the extended substrate 400-500 μm past the point of the electrode?

The custom Intan headstage only seems to have 64 contact pads available for making the connection between the board and Intan, how was the connection made to all electrodes?

What is the weight of the Titanium head cap when assembled and how does it compare to more standard caps?

Some small details about the implantation are still unclear to me. I believe the microwires are selected one at a time, the tungsten wire remains fixated to the bundle of electrodes due to the silk 'glue' as well during manipulation? The authors suggest that multiple boards/devices could be stacked to increase recording channel count / brain region access – would this be practically possible with the method used here to select the tungsten wires one by one when multiple bundles would essentially be on top of each other?

The hypothesis of a “cell-attached” is interesting – were impedance measurements carried out after implantation for this example? Is there any indication by the impedance value that this electrode may indeed be in more direct contact or ‘sealed’?

Is there a reason or meaning behind the abrupt increase in the Unit yield / recording contact of Figure 4d?

Regarding the immunohistology - were any horizontal plane brain slices made to better see the ‘footprint’ of the device along the trajectory of implantation?

The sudden year-long time scale of unit tracked in Figure 4 is confusing at first glance. Perhaps the figure could be updated to state clearly in part f) and in the figure caption that it is for mice.

What was the reason behind the chosen timeline of 3.5 months for the rat experiments?

The difference of filled and non-filled dots signifying members/non-members of ensembles, respectively, is extremely difficult to see, at least in the figure resolution of the received document.

Is the statement that the UTFEs achieve SNR of spikes 1.5 to 3 times higher than other technologies based on the highest values observed (~60 – 89)? How many recording sites achieved these large SNR values and is this a fair comparison/statement?

Are some electrodes “buried” inside the bundle and not capable of recording activity? If so, what is the percentage of channels that are lost due to this?

The discussion seems to hint that other studies require a stiff/invasive shuttle or wire to implant flexible probes, while the UTFE does not. This should be made clear as the developed implantation strategy uses wires of similar scale to some previous studies.

Although the manuscript is well-written, some improvement or polishing would be beneficial. Some language mistakes/misspellings are present and some sentences are written in a more conversational manner, in particular in the discussion.

Minor comments:

In general many of the figures are quite complex and have a lot of information/data. The authors have done a good job of separating and explaining this, however perhaps, for example, in Figure 3 the subheadings such as Dorsal hippocampus and Intermediate Hippocampus could also be underlined to more easily find the parts of the captions that correspond to the a), b), c), etc.

Do all parts of Table S1 other than the top row come from the current study?

SNR redefined on line 272

Reviewer #5 (Remarks to the Author):

General comments

The authors demonstrate a flexible polyimide based neural probe that enables the long-term recordings of neurons from multiple areas of the brain. The advances on state-of-the-art are primarily implant geometry that is enabled by complementary techniques for surgical implantation. The techniques presented within the paper are in line with current trends of making neural probes flexible and smaller to mitigate foreign body responses and neuronal damage. The results are noteworthy and will be of significant interest to the field.

Revisions

In the abstract, and in the results, provide a clear definition/explanation of ensembles so the data can be more widely understood (details should remain in the methods).

Can the authors include further discussion on the implications of this technological advancement.

Please clarify these 2 sentences in the abstract which aren't clear: 'The average ensemble lifetimes were shorter than the durations we can track individual cells (up to a year). Several ensembles remained inactive and reemerged only after several days/weeks despite UFTEs were able to track the firing of individual neurons within the ensembles uninterrupted.'

Abstract, 'the ensemble members were mostly hippocampal' – this statement cannot be appreciated as the reader hasn't been informed of the locations of the electrodes.

UFTEs can be inserted 6.5 mm deep. Can the authors add some discussion on how this could be increased to make this technology relevant to larger models and humans.

The authors do not sufficiently justify the claim of year-long stability. The year-long claim appears to come from data up to 10 months in mice, which indicates that only two single units were detectable up to 10 months. The ephys data, regarding SWRs and neuronal ensembles, impedance, and the IHC, used to highlight the absence of tissue damage, was only conducted in rats up to 3.5 months after implantation.

Throughout the text the authors should indicate species data relates to.

The statement "The average ensemble lifetimes were shorter than the durations we can track individual cells (up to a year)" conflates rat and mouse data. Figure 4e indicates that 20% of single units can be tracked up to 3 months, but the trend suggests a negligible amount would be tracked up to a year. Fig 4f, showing single units up to 10 months in mice, appears to be an outlier. The authors should provide a distribution of how long single units can be tracked in mice c.f. Fig 4e. This is important to justify the claim of year long stability.

Figure 3a suggests zero damage to any neuronal structures from the UFTE and the abstract states "Immunostaining revealed no detectable tissue damage even after several months." The reviewer appreciates that damage to surrounding neuron may be minimal, compared to existing probes.

However, this is only within the limitation of the staining/microscopy that was performed. It is likely the probes still cause some damage to fine neuronal structures on insertion.

Regarding in vivo impedance measurements, could the authors clarify the magnitude of the stimulation waveform, and the location and relative size of the counter electrode. Figure 4c should specify the frequency the impedance was measured at. Changes to the electrode-tissue interface are not exclusively represented by the Bode magnitude plot, or the 1 kHz impedance magnitude (assuming this is represented in 4c). The authors should also provide the phase plot corresponding to Fig 2B and similarly provide both Bode plots for characteristic electrodes at 3 months in vivo.

Ensembles include two hippocampal regions and retrosplenial and prefrontal cortex all thought to be related in spatial navigation, memory and memory consolidation. Interpretation of this requires more detailed descriptions of what the animals are doing during recording.

The authors should add details of:

- Where the rats are recorded, what environment, is it a familiar or novel environments, is it the animals homecage - if so, are they recorded in the absence of cagemates.
- Is it an empty environment, how large?
- Are they plugged in to a wired headstage?
- How frequently recordings were taken

This info should be provided in detail in methods. It should also be briefly mentioned in main text. "Rats were recorded daily for X time in an x metre box".

Could the authors comment on:

- The lateral spread of the electrodes
- The weight of the assembled headstage
- The strain, sex, age and weight at surgery

RESPONSE TO THE REVIEWERS

We would like to thank all three reviewers for very helpful and constructive feedback. Please find below our point-by-point reply to the reviewers' comments. We highlighted all changes in the manuscript and the supplementary information in yellow (some minor stylistic changes, particularly in the Introduction and Discussion, in accordance with Comment 2.16, were not highlighted for better readability). We uploaded the revised manuscript text and the supplementary information to the Manuscript Tracking System. In addition to the changes in Figures asked by the reviewers, we made minor modifications in Part ii of the "Intermediate Hippocampus" panel and the "Single units from mPFC" panel of Figure 3 according to the latest version of the spike sorting results we included in the other parts of the manuscript; in the panels b and f of Figure 4 to correct for minor errors in the scaling and spatial organization of the mean spike waveforms; in panel d of Figure 4 and the corresponding text in Results to correct for a minor error in the distribution of cortical and hippocampal units, and in the axes of panel f of Figure 5 to make them more consistent with the main text. Since the submission of our manuscript, we have also successfully tested a 512-channel wireless logger, which we now mention (The long-term ensemble data in the manuscript were taken using our previous generation 256-channel wired and wireless loggers).

We also addressed the editorial requests by uploading all the data presented in Figures and Supplementary Figures in separate Excel sheets, by uploading separate PDF files for each main figure, and by uploading a Supplementary Information PDF file that includes all the supplementary figures and tables to the Manuscript Tracking System.

Reviewer #1:

Comment 1.1: *Yasar et al. describe a novel electrode design for extracellular electrophysiological recordings consisting of ultraflexible electrode bundles and demonstrate remarkably stable recordings in rats over multiple months using this design. The authors are to be congratulated for a very impressive proof-of-concept manuscript on a tool that could be useful for a wide range of neuroscience researchers. The paper is well written and without major methodological flaws. The results regarding the long-term stability of assemblies are potentially interesting, although some caution is warranted given that they are based on two animals only. I have two major concerns regarding the statistical analyses and the description of the methods used. Neither of these would alter the main conclusions of the paper, but in my opinion they should be remedied before the paper can be accepted. There are also some minor glitches and ambiguities that could be improved in a revised version, as outlined below.*

Response 1.1: We would like to thank the reviewer for the very helpful feedback and critique. We have addressed the improvements suggested by the reviewer below:

Major comment 1.2: *The description of the algorithms used for spike sorting are conflicting between the method's section and the main text. The text references Jun et al., Biorxiv and correctly describes the process underlying Kilosort, though it is unclear which version of the algorithm was used. The methods refer to JRCLUST and Tank et al. as a reference, which seems to be based on multi-unit activity. Please clarify which algorithm and version was used in the methods and cite the corresponding paper.*

Response 1.2: Thank you for catching the typo in the JRCLUST citation in the Methods section. We fixed it now to cite the Jun et al. 2017 Biorxiv preprint (DOI: 10.1101/101030). This preprint describes the process underlying JRCLUST, which we used in our study. In the Methods section, we also now specified the version of the JRCLUST that was used.

Major comment 1.3: *The statistical analysis used in Fig 4c (Wilcoxon Rank sum test based on arbitrarily chosen time points) is inappropriate in my opinion. First, the data points represent repeated measures of impedances in the same channels and paired testing would be indicated. Second, if individual time points are picked for comparison, some justification as to how they were chosen should be given and they should be consistent between the two animals (e.g. week 1 vs. week 4, week 4 vs. week 12 or similar). The correct form of statistical test to show that impedances do not change would involve some form of equivalency testing for time series.*

On the other hand, I don't think that statistical testing is strictly necessary to begin with as the figure merely describes the stability of impedances over time and the relevant information is clear without p-values.

Response 1.3: Per the reviewer's suggestion, we removed the statistical testing and only reported the impedance values for each rat across months in the text.

Minor comment 1.4: Line 63, Introduction: The authors should cite Schoonover et al., 2020, Nature, PMID 34108681 as a counterexample that used conventional silicon probes over multiple weeks with highly optimized surgical techniques. They may want to discuss this reference at some point later, too.

Response 1.4: We now cited the suggested study using stiff silicon electrodes with up to 1-2 months long recordings in the Introduction and discussed it in the Discussion section.

Minor comment 1.5: Line 189: The abbreviations IL, PrL and Cg1 should be introduced here (or somewhere else) as they are later used in multiple figure legends without ever being introduced.

Response 1.5: Per the reviewer's suggestion, we introduced the abbreviations IL, PrL, and Cg1 in Line 181 (which corresponds to Line 189 in the original submission).

Minor comment 1.6: Line 200: it is not entirely clear here or in the method what the earliest time point after surgery was at which the authors attempted to record. Figure 4c indicates that impedances may have been very low shortly after implantation. Therefore, this information would be interesting and should be added either here or in the methods.

Response 1.6: After UFTEs were implanted, we waited at least two days and a maximum of six days to start the first recordings, which was the post-surgery recovery period according to our animal experiment license from Cantonal authorities. The in vivo impedances measured at 1 kHz (Fig. 4c) started higher than the in vitro values (Fig. 2b), most likely due to the more restricted access of the ionic currents through the biological tissue compared to saline. Afterwards, the impedances very slightly increased during the following weeks until they stabilized. We clarified these now in the Results section of the manuscript.

Minor comment 1.7: Line 709: states the craniotomy for inserting the probe was made over the cerebellum but the text states that recordings in mice were made in CA1. Please clarify / correct as appropriate.

Response 1.7: Thank you for catching the typo; we corrected "cerebellum" to "cerebrum." Furthermore, we have also included the exact stereotaxic coordinates of the mouse craniotomies now in the corresponding part of the Methods section.

Minor comment 1.8: Fig 2a (v): Would recommend a higher magnification and higher resolution for this photograph, otherwise its hard to take anything away from it.

Response 1.8: We have replaced this image with a higher magnification and resolution version. We included a larger version of this image in the supplementary (Fig. S8).

Reviewer #2:

Comment 2.1: Yasar et al. report on the development of a 256-channel flexible recording array with 4 bundles of physical separated electrodes capable of recording over multiple months. The channel count is impressive, in particular for a probe that makes good use of the material flexibility to avoid adverse reaction of the tissue to the implant. Additionally, the ability to probe 4 different brain regions is interesting and the developed method of implantation to enable choice of brain regions and implant angles is valuable. The paper is well written, the technology and data are presented well, and the demonstrated capability to record over months is significant. I have some questions/comments, but overall, I find that the manuscript is already in a good state.

Response 2.1: We would like to thank the reviewer for the very useful feedback and critique that helped improve the manuscript.

Comment 2.2: Page 3, Line 100 – does the comparison stating the mesh electrode has 22 larger cross-sectional area per recording contact include the "empty space" of the mesh? If so, is this a fair comparison?

Response 2.2: According to the methodology describing the mesh electrode (Zhao et al. 2023, Nature Neuroscience, PMID 36804648), the "empty space" is filled with polyethylene glycol and, therefore, not empty during the electrode insertion (for fair comparison, we also account for such insertion glue/shuttle dimensions for our probes too). Since we wanted to compare the acute damage caused by the insertion of the different types of electrodes, we are referring to the mesh electrode's cross-sectional area, including the "empty space." We now emphasize this more clearly in the Introduction.

Comment 2.3: *What is the reason for the extended substrate 400-500 μm past the point of the electrode?*

Response 2.3: Fibers being terminated by a thin fiber extension results in a more streamlined conical shape on the tips of the UFTE bundles. This, in turn, helps the insertion of the UFTE bundles into the brain with a reduced risk of electrode fibers separating from each other prematurely before reaching the targeted depth in the brain. We have now explained this rationale in the Results section.

Comment 2.4: *The custom Intan headstage only seems to have 64 contact pads available for making the connection between the board and Intan, how was the connection made to all electrodes?*

Response 2.4: Supplementary Figure 2A (left) illustrates that we are using 4 x RHD2164 64-channel amplifier chips (4x64=256 channels) on the top side of the head-stage PCB. Supplementary Figure 2A (right) shows the bottom side of the head stage, revealing the presence of 256 contact pads on the board, along with a transparent surface-mounted flexible electrode. We have now further clarified this in the Methods section.

Comment 2.5: *What is the weight of the Titanium head cap when assembled, and how does it compare to more standard caps?*

Response 2.5: The base is 0.25 g (the only part cemented to the skull), the right side of the shell is 8g, the left is 5g, and the head stage is 0.45 g. The total mass during recording is 13.7g. In the home cage, when a top part covers the head stage, this also adds eight more grams to the total mass of the TitaniumHelmet. We now also added the weights of different parts of the TitaniumHelmet in the methods section.

Headley et al. (<https://doi.org/10.1152/jn.00955.2014>) developed a cap for rats with a total weight of 16.9 g made from Acrylonitrile butadiene styrene (ABS)/polylactic acid (PLA) plastic, which is not biocompatible. According to the author, this biocompatibility problem was mitigated because dental acrylic was placed between the 3D-printed components and the skull. There is also a lighter rat cap developed by Vöröslakos et al. (<https://doi.org/10.7554/eLife.65859>), which is only 8.3 g and made from clear v4 (RS-F2-GPCL-04, Formlabs) resin which is not biocompatible. Vöröslakos' cap has a smaller area for craniotomies with a fragile wall and requires unscrewing to open the cap. Our goal was to design a head cap that is (1) fully biocompatible (medical grade titanium), (2) easy to open (held by magnets), and (3) that provides protection and stability for a head stage throughout the animal's lifetime without requiring single housing of rats (by withstanding chewing by cage-mates). We have now added references and emphasized these challenges in the methods section.

Comment 2.6: *Some small details about the implantation are still unclear to me. I believe the microwires are selected one at a time, the tungsten wire remains fixated to the bundle of electrodes due to the silk 'glue' as well during manipulation? The authors suggest that multiple boards/devices could be stacked to increase recording channel count / brain region access – would this be practically possible with the method used here to select the tungsten wires one by one when multiple bundles would essentially be on top of each other?*

Response 2.6: The silk glue holds only the electrode fibers together for each bundle but is not used to connect the bundle to the tungsten. The bundle of electrode fibers is tethered to the tungsten wire purely through the mechanical engagement of the loop at the end of the longest fiber in the bundle with the tip of the tungsten wire during manipulation/implantation. The tungsten wire and electrode bundle assemblies are picked for insertion one at a time during the implantation.

There is a predefined implantation sequence from the anterior to posterior bundles, with the most anterior bundle being the first to be implanted. If two bundles were implanted next to each other in the same coronal plane of the brain, the more medial one precedes the more lateral one.

In the case of multiple headstage boards, one can implant bundles in the sequence described above, starting with the first headstage board (bottom of the stack). When the bundles connected to the first board are implanted, the headstage from the stereotaxic arm can be released (with the same male connector on the bottom side of each headstage) and temporarily held on the side with a holder. The second headstage, loaded with another group of UFTE bundles, can then be attached to the stereotaxic arm. The new bundles can be implanted either in the same hemisphere as the previous bundles by following the same anterior-posterior sequence as before or in the other hemisphere. Afterwards, the second headstage board can then be plugged into the first one by sliding the uppermost top headstage board into its respective slot inside the TitaniumHelmet. The shell of TitaniumHelmet only holds the top headstage because this is exposed to make a connection with the recording hardware.

We now extended the Methods section with all the details mentioned above.

Comment 2.7: *The hypothesis of a “cell-attached” is interesting – were impedance measurements carried out after implantation for this example? Is there any indication by the impedance value that this electrode may indeed be in more direct contact or ‘sealed’?*

Response 2.7: Yes, the impedance measurement on the electrode site with the “cell-attached” recording resulted in an impedance magnitude of 7.88 M Ω , which is significantly higher than the impedance magnitudes of other electrode sites where single-unit activity is typically recorded (300 k Ω -1 M Ω). Furthermore, this impedance magnitude is even higher than the ones of solely gold electrode sites of equal size without any PEDOT:PSS coating (approximately 2-3 M Ω). Such a high impedance magnitude could indicate a restricted flow of ions around the electrode site, further supporting the idea that the recorded neuron partially sealed the electrode site. We now further elaborated on this in the Discussion.

Comment 2.8: *Is there a reason or meaning behind the abrupt increase in the Unit yield / recording contact of Figure 4d?*

Response 2.8: The abrupt increase from Week 12 to Week 13 is caused by the data point on Week 13 because Weeks 0 and 13 consist of data from only one rat (there was a one-week offset between the post-surgery recording periods of the two rats). For the sake of consistency between sample sizes behind each data point, we now present data from only Week 1 to Week 12 in Fig. 4d, where each point has data from both rats.

Comment 2.9: *Regarding the immunohistology - were any horizontal plane brain slices made to better see the ‘footprint’ of the device along the trajectory of implantation?*

Response 2.9: We chose a coronal slicing to see an electrode track as it is on the **same plane** to quantify the number of the glial cells and activated astrocytes around the electrode bundle. Due to the small footprint of the implant and the slight deviations in insertion angles, it was a highly challenging process for us to capture a single coronal slice (from both rat brains we recorded) that shows the entire electrode tracks. This slicing process was not compatible for us to also get any horizontal slices.

Comment 2.10: *The sudden year-long time scale of unit tracked in Figure 4 is confusing at first glance. Perhaps the figure could be updated to state clearly in part f) and in the figure caption that it is for mice.*

Response 2.10: We now clarified in the title of Fig. 4f that the data is from the mouse hippocampus. Please also see our Response 5.7 to Reviewer #5 regarding mice vs rat recordings and clarifications we made.

Comment 2.11: *What was the reason behind the chosen timeline of 3.5 months for the rat experiments?*

Response 2.11: The animal research protocol approved by the cantonal veterinary office restricted the rat experiments conducted in this study to 14 weeks.

Comment 2.12: *The difference of filled and non-filled dots signifying members/non-members of ensembles, respectively, is extremely difficult to see, at least in the figure resolution of the received document.*

Response 2.12: Figure 5a was modified based on the reviewer's suggestion. The filled dots at the end of the neuron ensemble weights are now enlarged, increasing their visibility compared to the unfilled dots (indicating non-members of the ensemble).

Comment 2.13: *Is the statement that the UTFEs achieve SNR of spikes 1.5 to 3 times higher than other technologies based on the highest values observed (~60 – 89)? How many recording sites achieved these large SNR values and is this a fair comparison/statement?*

Response 2.13: The SNR comparison is based on the **mean** SNR of all single units we recorded across the sessions vs. the **mean** SNRs of the single units recorded by other state-of-the-art ultra-flexible electrode arrays (i.e., Z. Zhao et al., 2022 PMID: 36192597 and S. Zhao et al., 2023 PMID: 36804648). The mean SNRs used in this comparison come from the data provided for Figure 6h of S. Zhao et al., 2023 paper (**8.96 ± 0.39, mean ± s.e.m.**) and the mean SNR of **5.51** reported in the Results section of the Z. Zhao et al., 2022 paper. In comparison, the mean SNR of the spikes of all single units (also averaged across sessions during which the unit was tracked) we recorded is **14.6 ± 0.5 (mean ± s.e.m.)**, which is approximately 1.5-3x of these previous studies. On the other hand, the highest mean SNRs we observed were **6 to 20 times higher** than the mean SNR values previously reported. However, we did not base our claims on these ones. Now, we have separately emphasized the mean and highest SNRs in the manuscript.

Comment 2.14: *Are some electrodes “buried” inside the bundle and not capable of recording activity? If so, what is the percentage of channels that are lost due to this?*

Response 2.14: While this is possible, we have not observed a significant loss of recording contacts due to such a scenario. The percentage of broken channels in the in vitro impedance measurements was approximately 1.6%, which can be attributed to defects in the microfabrication or soldering processes. The percentage of “lost” recording contacts that we excluded from our spike sorting analysis due to not recording any local field potential or spiking activity was 3 to 6%. In addition to the recording contacts being buried inside the bundle, the additional 1.5 to 4.5% lost recording contacts observed in vivo can also be attributed to any damages to the UFTE fibers or the ribbon cables connecting the UFTE bundles to the headstage board during the electrode assembly or implantation. We now mention this in the manuscript (Results).

Comment 2.15: *The discussion seems to hint that other studies require a stiff/invasive shuttle or wire to implant flexible probes, while the UTFE does not. This should be made clear as the developed implantation strategy uses wires of similar scale to some previous studies.*

Response 2.15: UTFEs also rely on tethering to a stiff shuttle for insertion. We clarified the wording in the last sentence of the paragraph in the Discussion where we compare our technology to other flexible/ultra-flexible electrode technologies to state this point more clearly.

Comment 2.16: *Although the manuscript is well-written, some improvement or polishing would be beneficial. Some language mistakes/misspellings are present and some sentences are written in a more conversational manner, in particular in the discussion.*

Response 2.16: We went over the manuscript several times again, and asked other colleagues also to comment. We will greatly appreciate it if the reviewer still sees any further phrases/sentences to be changed; he/she can point out those instances.

Comment 2.17: *In general, many of the figures are quite complex and have a lot of information/data. The authors have done a good job of separating and explaining this, however perhaps, for example, in Figure 3, the subheadings such as Dorsal hippocampus and Intermediate Hippocampus could also be underlined to more easily find the parts of the captions that correspond to the a), b), c), etc.*

Response 2.17: As suggested, we made all the subheadings in Figures 3 and 4 bold and increased their font size to make them more distinct in the figure.

Comment 2.18: *Do all parts of Table S1 other than the top row come from the current study?*

Response 2.18: We collected all the data presented in all rows of Supplementary Table 1 under the current study. These data are from electrode insertions into 0.9% agar (in vitro), whereas the data presented in the main text are from insertions into the rat or mouse brain (in vivo). We now clarified this.

Comment 2.19: SNR redefined on line 272

Response 2.19: Thank you, we removed the redundant definition on Line 258 (Line 272 in the original submission).

Reviewer #5:

Comment 5.1: *The authors demonstrate a flexible polyimide based neural probe that enables the long-term recordings of neurons from multiple areas of the brain. The advances on state-of-the-art are primarily implant geometry that is enabled by complementary techniques for surgical implantation. The techniques presented within the paper are in line with current trends of making neural probes flexible and smaller to mitigate foreign body responses and neuronal damage. The results are noteworthy and will be of significant interest to the field.*

Response 5.1: We would like to thank the reviewer for the very helpful feedback and points.

Comment 5.2: *In the abstract, and in the results, provide a clear definition/explanation of ensembles so the data can be more widely understood (details should remain in the methods).*

Response 5.2: The definition of neuronal ensembles has been added to the Abstract and Results sections.

Comment 5.3: *Can the authors include further discussion on the implications of this technological advancement.*

Response 5.3: We now added one short paragraph at the end of the Discussion section on the potential use and impact of UTFEs in basic research and clinical use. We also added one longer paragraph on the relevance of our findings on hippocampal ensembles to the previous research done in the field on cortico-hippocampal memories, plasticity, and neuronal types/connectivity.

Comment 5.4: *Please clarify these 2 sentences in the abstract which aren't clear: 'The average ensemble lifetimes were shorter than the durations we can track individual cells (up to a year). Several ensembles remained inactive and reemerged only after several days/weeks despite UTFEs were able to track the firing of individual neurons within the ensembles uninterrupted.'*

Response 5.4: We have now modified the Abstract to make it more readable. Due to the space limitation in the Abstract, we elaborated on this statement further in the Results section (yellow highlighted in the paragraph before the last one). The ensemble lifetimes were significantly lower than the duration where the neurons in the ensemble were detectable. Thus, the ensemble lifetime measurements were not limited due to our ability to track the individual neurons within the ensembles.

The ensembles in rats were tracked for three and a half months. We also made the necessary corrections to avoid any ambiguities with respect to the almost-year-long mouse data.

Comment 5.5: *Abstract, 'the ensemble members were mostly hippocampal' – this statement cannot be appreciated as the reader hasn't been informed of the locations of the electrodes.*

Response 5.5: Thank you, we now modified the Abstract to include the brain areas.

Comment 5.6: *UTFEs can be inserted 6.5 mm deep. Can the authors add some discussion on how this could be increased to make this technology relevant to larger models and humans.*

Response 5.6: The electrode implantation technology introduced in this study has no inherent limit on insertion depth (to the best of our knowledge and experience so far). The 3x silk fibroin coating on the electrode bundle lasts at least several hours, which is beyond the duration of the mechanical insertion of UFTE bundles into any possible target area in larger animal models or humans. Yet, this long dissolution time

does not limit us from retracting the stiff shuttles because they are only mechanically coupled to UFTE bundles. Our electrode insertion method can be realized with any other biodegradable coating material with a sufficiently long dissolution time scale.

On the other hand, the tungsten shuttles we used in the current study, which had a 50- μm diameter, may not have sufficient stiffness to reach deep brain targets accurately in large animal models or humans. In that case, these shuttles can be replaced by shuttles with larger diameters and/or materials with higher Young's modulus. Nevertheless, these are modifications that do not require deviations from our methodology. We included a summary of these points as a paragraph in the Discussion (highlighted in yellow).

Comment 5.7: *The authors do not sufficiently justify the claim of year-long stability. The year-long claim appears to come from data up to 10 months in mice, which indicates that only two single units were detectable up to 10 months. The ephys data, regarding SWRs and neuronal ensembles, impedance, and the IHC, used to highlight the absence of tissue damage, was only conducted in rats up to 3.5 months after implantation.*

Response 5.7:

Thank you, we have now clarified the recordings at various points in the manuscript:

2 mice up to 10 months: We had only 2 mice that were recorded up to 10 months (by Helmchen lab) due to the restrictions of animal experiment durations in protocols in Switzerland. We used data from only one mouse for a complete single-unit tracking analysis because the interval between recordings was less regular in these long mouse recordings, which limited the analysis. We now added the plots for the SNR stability, the distribution of the tracking duration of single units in the mouse hippocampus, and the electrode impedance stability in Supplementary Figure 9. UFTEs provided high-quality single-unit activity for almost a year (i.e., 10 months) in the mouse brain, regardless of how many of the single units were tracked for up to a year (and even though the recording contacts were tetrodes, i.e., 4 contacts per fiber, with larger surface area and arguably less biocompatibility than the UFTE fibers we used for the majority of the data in the manuscript). We also changed everywhere “a year” to “almost a year” and indicated 10 months as the longest duration tested. There is nothing special about the 10-month duration besides the unfortunate Cantonal regulatory restrictions on the duration of experiments in our mouse licenses. IHC was not conducted in the brains of these mice for microglia and astrocytes since they were implanted for a different project in the Helmchen lab.

2 rats up to 3.5 months: All the rat studies were conducted in our lab under a license that permits experiments up to 3.5 months. Here, we had the full permit/opportunity to do the multi-areal ensemble tracking and sharp-wave analysis, as well as the extensive tissue processing with IHC.

Comment 5.8: *Throughout the text the authors should indicate species data relates to.*

Response 5.8: We added the species information in the places where it was not clearly indicated for each main figure and each paragraph in the Results section.

Comment 5.9: *The statement “The average ensemble lifetimes were shorter than the durations we can track individual cells (up to a year)” conflates rat and mouse data. Figure 4e indicates that 20% of single units can be tracked up to 3 months, but the trend suggests a negligible amount would be tracked up to a year. Fig 4f, showing single units up to 10 months in mice, appears to be an outlier. The authors should provide a distribution of how long single units can be tracked in mice c.f. Fig 4e. This is important to justify the claim of year long stability.*

Response 5.9:

Thank you, we now removed the “(up to a year)” from the statement in the Abstract in order to avoid any ambiguity. The results based on Figure 5 were exclusively recorded from rats. We now added the distribution of the tracking duration of single units in the mice hippocampus in Supplementary Figure 9. In fact, this distribution also follows the trend in Fig. 4e.

The ensemble lifetimes on average (rat data) were shorter than the durations we can track individual neurons of the ensembles (rat data). Thus, ensemble detectability and lifetimes measured were not significantly affected by the trackability of individual neurons.

Comment 5.10: *Figure 3a suggests zero damage to any neuronal structures from the UFTE and the abstract states “Immunostaining revealed no detectable tissue damage even after several months.” The reviewer appreciates that damage to surrounding neuron may be minimal, compared to existing probes. However, this is only within the limitation of the staining/microscopy that was performed. It is likely the probes still cause some damage to fine neuronal structures on insertion.*

Response 5.10:

- (a) Because of “the limitation of the staining/microscopy,” as the reviewer pointed out, we particularly use the word “detectable” in the abstract. We now also went through the manuscript and added “detectable” everywhere related.
- (b) Our statement (based on the immunostaining results presented in Fig. 4g) is strictly for the long-term tissue response (the tissue processing and immunostaining were done well beyond the typical timescales for tissue healing after acute insertion damage). It is certainly true that the initial insertion can cause more damage, which is also visible from the slight recovery/improvement in impedances and SNRs. Therefore, we now added the word “chronic” to the tissue damage claims in the Results and Abstract to further clarify this point. The chronic tissue damage is the eventual point that matters for most purposes and is the one reported by the papers in the field.

Comment 5.11: *Regarding in vivo impedance measurements, could the authors clarify the magnitude of the stimulation waveform, and the location and relative size of the counter electrode. Figure 4c should specify the frequency the impedance was measured at. Changes to the electrode-tissue interface are not exclusively represented by the Bode magnitude plot, or the 1 kHz impedance magnitude (assuming this is represented in 4c). The authors should also provide the phase plot corresponding to Fig 2B and similarly provide both Bode plots for characteristic electrodes at 3 months in vivo.*

Response 5.11: Both the in vivo (Fig. 4c) and in vitro (Fig. 2b) impedance measurements were conducted by the Intan RHD USB interface board. According to the datasheets of the RHD2164 chip and RHX Acquisition Software and the source code provided by Intan Technologies, a sinusoidal current wave of the desired test frequency was generated by coupling a digital-to-analog converter (DAC) to a capacitor (0.1 pF, 1 pF, or 10 pF) and injected into the electrode. The corresponding amplifier input channel measured the resulting voltage across the electrode and the tissue during the impedance measurement. The voltage generated by the DAC and the capacitor connected to the DAC was selected as a result of an algorithm that tried to keep the voltage measured by the amplifier around 250 μ V, which, according to Intan, gives the most accurate impedance measurements without reaching the saturation (at ± 5 mV).

All impedance measurements were performed in a two-electrode setup. In the in vitro impedance measurements, the reference/counter electrode was an Ag/AgCl wire also immersed in the saline along the electrode arrays. In the in vivo impedance measurements, the reference/counter electrode was the 0.9 mm-diameter stainless steel screw placed on the cerebellum (approximate coordinates: -12.5 mm AP, 2.5 mm ML), which also served as the reference during the recordings. We have now added some of these details in the Methods section.

We now clarified the impedance measurement frequency (1 kHz) in the caption of Fig. 4c and the paragraph corresponding to Fig. 4c in the Results section. We added the phase plot corresponding to the in vitro impedance measurements at different frequencies (Fig. 2b) as a supplementary figure (Supplementary Fig. 7). We also added the statistical summary of the impedance phases corresponding to the measurements at 1 kHz (Fig. 4c) in the Supplementary Fig. 7 as well. It can be seen from this figure that the resistive/capacitive nature of the electrode-tissue interaction remains stable across 3 months. While the phases are mostly between -10 and -20 degrees, which would normally indicate a resistive interaction between the recording contacts and the tissue, one should approach these values with caution since the ionic currents generated in the tissue can act as non-linear elements in the overall circuit model and interfere with the magnitude and the phase of the impedance. Unfortunately, we did not measure the impedances across the entire frequency spectrum in vivo since such measurements take a long time for all 256 channels (approximately 20-30

minutes). This would have created an additional burden on our rats and was not covered by the Cantonal animal experiment license.

Comment 5.12: *Ensembles include two hippocampal regions and retrosplenial and prefrontal cortex all thought to be related in spatial navigation, memory and memory consolidation. Interpretation of this requires more detailed descriptions of what the animals are doing during recording. The authors should add details of:*

- *Where the rats are recorded, what environment, is it a familiar or novel environments, is it the animals homecage - if so, are they recorded in the absence of cagemates.*
- *Is it an empty environment, how large?*
- *Are they plugged in to a wired headstage?*
- *How frequently recordings were taken*

This info should be provided in detail in methods. It should also be briefly mentioned in main text. "Rats were recorded daily for X time in an x metre box".

Response 5.12:

Thank you, we have now added all missing information to the Methods and referenced in the Results:

All the multi-areal recordings and ensemble analyses were done in rats. The rats were familiar with the environment (A 50x50x50cm plexiglass cage covered by copper mesh on the sides and bottom but opened on the top side). The cage's floor was covered with bedding and changed after each recording session, which was conducted twice a week. Within the cage, a glass petri dish with a single drop of concentrated milk was a positive reward after connecting the recording system to the rat. We uploaded a video as a manuscript supplement demonstrating the head direction cell. This video offers a glimpse into the exact conditions of the rats during the wired recording.

The custom-made head stage (see Supplementary Figures 2) was enclosed within a TitaniumHelmet, and the rats lived with the head stage throughout the entire experiment. We utilized a small PCB with a connector for digital signals only, plugged into the encapsulated head-stage after opening the magnet-held cover of the TitaniumHelmet. Both tethered and untethered (wireless) experiments were conducted on the same animals (see Supplementary Figures 3 C-D).

(Recordings from mice were performed in head-fixed mice during spontaneous behavior on a linear treadmill. Mice were free to walk on the treadmill or remain stationary and had no additional sensory stimulation or reward delivery.)

Comment 5.13: *Could the authors comment on:*

- *The lateral spread of the electrodes*
- *The weight of the assembled headstage*
- *The strain, sex, age and weight at surgery*

Response 5.13:

- According to the state of the electrode bundles in the explanted and processed brain tissue, the lateral spread of the electrode fibers in a bundle was approximately 100 μm . An example image is shown in Figure 1d. We now point this out in the manuscript.
- The headstage weighs 0.45 g. The weight of the UFTes, which consist of 2.4- μm -thick polyimide films, is negligible. Please also see our detailed response to Reviewer #2, Comment 2.5. We have now added detailed information in the Methods.
- The rats used in this study were all female Long Evans rats. The ages of rats at surgery ranged between 23 and 59 weeks, and their weights ranged between 270 and 340 g. We added this information to the Methods section as well as details of the mice used in the study.

REVIEWERS' COMMENTS

Reviewer #5 (Remarks to the Author):

The authors have thoroughly addressed all comments and made suitable changes to the manuscript. The reviewer recommends publication.